# A Phase-Field Model for In-Space Manufacturing of Binary Alloys

**DOI:** 10.3390/ma16010383

**Published:** 2022-12-31

**Authors:** Manoj Ghosh, Muhannad Hendy, Jonathan Raush, Kasra Momeni

**Affiliations:** 1Department of Mechanical Engineering, Michigan State University, East Lansing, MI 48824, USA; 2Department of Mechanical Engineering, University of Alabama, Tuscaloosa, Al 35487, USA; 3Department of Mechanical Engineering, The University of Louisiana at Lafayette, Lafayette, LA 70503, USA

**Keywords:** additive manufacturing, phase transformation, diffuse interphase, partitionless solidification

## Abstract

The integrity of the final printed components is mostly dictated by the adhesion between the particles and phases that form upon solidification, which is a major problem in printing metallic parts using available In-Space Manufacturing (ISM) technologies based on the Fused Deposition Modeling (FDM) methodology. Understanding the melting/solidification process helps increase particle adherence and allows to produce components with greater mechanical integrity. We developed a phase-field model of solidification for binary alloys. The phase-field approach is unique in capturing the microstructure with computationally tractable costs. The developed phase-field model of solidification of binary alloys satisfies the stability conditions at all temperatures. The suggested model is tuned for Ni-Cu alloy feedstocks. We derived the Ginzburg-Landau equations governing the phase transformation kinetics and solved them analytically for the dilute solution. We calculated the concentration profile as a function of interface velocity for a one-dimensional steady-state diffuse interface neglecting elasticity and obtained the partition coefficient, k, as a function of interface velocity. Numerical simulations for the diluted solution are used to study the interface velocity as a function of undercooling for the classic sharp interface model, partitionless solidification, and thin interface.

## 1. Introduction

The further away from Earth, the more difficult it will be to transport all the supplies and redundant parts the astronauts may require. As a result, a strong ISM capability for sophisticated and lightweight materials is a must for such missions. NASA has successfully printed polymer parts on the International Space Station (ISS) using the Fused Deposition Modeling (FDM) technology. However, metallic component printing techniques for ISM are still in development [1]. Microgravity ISM cannot be made using typical powder-based additive manufacturing processes. The FDM process is the most promising technology for ISM printing of metallic parts. Printing objects with high integrity, which is directly related to the creation of surface melt for loosely packed metallic powders, is a significant technological difficulty associated with FDM.

The requirement for new high-fidelity analytical and computational tools to predict the process-microstructure-property correlation is one of the critical challenges in the reliable manufacturing of metallic parts utilizing additive manufacturing techniques. The phase-field approach is a powerful and versatile tool for simulating microstructure evolution at the mesoscale, and it has recently become a popular method for studying various microstructure evolutions. Internal variables called order parameters are used to describe the form and distribution of grains in microstructure. These order parameters remain constant within the grains. The narrow region, where the order parameters change among adjacent grains, is known as the interface. The change in order parameters gives the time-dependent evolution of the microstructure. Reduction in bulk free energy, interfacial energy, and elastic energy are some of the driving forces for microstructure evolution. 

In the phase-field method framework, generally, two continuum equations, known as the Cahn-Hilliard nonlinear diffusion equation [2,3] and the Allen-Cahn equation [4], describe the microstructure evolution. The WBM (Wheeler, Boettinger, and McFadden) phase-field model has also been introduced, which deals with the isothermal solidification of a binary alloy. In this model, free energy functional and field equations were developed for the two types of order parameters, i.e., conserved and nonconserved [5]. The WBM model considered both local and gradient free energy and was used to study the impact of solute trapping during rapid solidification [6]. A phase-field model for rapid solidification of the binary alloys that recovers sharp interface solution has also been developed exhibiting solute trapping by asymptotic analysis [7], which later was expanded to non-isothermal solidification of binary alloys [8,9]. Phase-field simulations for dendrite growth coupled with heat and solute diffusion have been presented for a thin interface [10,11], which is valid for unequal solutal diffusivities in the solid and liquid. 

Hyperspherical phase-field models for rapid solidification neglecting the surface energy inhomogeneities have recently been developed for diffusionless processes neglecting elasticity [12], with elasticity [13,14,15], and with elasticity and surface tension [16] that satisfy all stability conditions for a three-phase system. Multiphase-field models have been developed and utilized to study the microstructure of printed Inconel 718 alloy [17] and solute trapping behavior during rapid solidification [18]. 

Coupled non-equilibrium phase-field and finite element thermal models are used to investigate the microstructure evolution during laser powder bed fusion of Ni-Nb alloy [19]. To predict the solidification structure during the rapid solidification processes, both interface kinetics and thermal diffusion need to be considered [20]. Finite- [21] and thin-interface [22] phase-field models have been developed to study highly non-equilibrium solidification processes and solute trapping during additive manufacturing. A dilute binary alloy phase-field model [23] has also been developed that maps onto the sharp interface continuous growth model [24] for various kinetic effects like solute trapping and solute drag to study the microstructure maps of rapid solidification. The phase-field model was combined with thermal and solutal diffusion as well as solute trapping effect to predict the microstructure for rapid solidification of dilute binary alloy. Phase-field models can also be used to investigate the sensitivity of the final manufactured parts to the variation in manufacturing conditions [25,26]. Recent progress in the development of phase-field models capturing the microstructure of printed parts is reviewed in Refs. [27,28,29,30,31,32].

In this study, we develop a phase-field potential for binary alloys that satisfies the stability conditions at all temperatures by capitalizing on our models for diffusionless melting/solidification [12,15,33] and materials growth [34,35]. We analytically solved the governing equations for dilute solution approximation and calculated interface velocity as a function of undercooling. We also derived the phase-field model for the thin interface limit and investigated the solute trapping for different interface velocities. Furthermore, we studied the effect of diffusivity and nonlocal energy on the equilibrium composition and the interface velocity. The proposed model, although limited to binary alloys and processes involving near-equilibrium thermodynamics, such as the fused deposition modeling, provide the basis for the development of more advanced models taking into account complex alloys and additive manufacturing processes involving far from equilibrium processes.

## 2. Phase-Field Model

The phase-field approach can predict the microstructure evolution using a set of internal conserved and nonconserved field variables, continuous along with the interface. These variables control the total free energy of the inhomogeneous microstructure system. To minimize the system’s total energy, following the second law of thermodynamics and assuming a linear relationship between the rate of change of order parameters and thermodynamic driving forces, we derive the Ginzburg-Landau (GL) kinetic equations governing the evolution of microstructure. The Helmholtz free energy of the system can be defined as,
(1)$$\psi =\psi^{l}+\psi^{\nabla }=\psi^{\theta }+{\overset{˘}{\psi }}^{\theta }+\psi^{\nabla },$$

Here, $\psi^{l}$ is the local free energy and $\psi^{\nabla }$ is the gradient energy. The local free energy is expressed as the sum of thermal energy $\psi^{\theta }$ and the chemical double-well potential energy $\;{\overset{˘}{\psi }}^{\theta }$,
(2)$$\psi^{\theta }\left(\theta ,c,\Upsilon \right)=G_{0}^{\theta }+\Delta G_{S0}^{\theta }\left(\theta ,c\right)q\left(\Upsilon ,a\right),$$
(3)$${\overset{˘}{\psi }}^{\theta }\left(\theta ,c,\Upsilon \right)=A^{S0}\left(\theta ,c\right)\overset{˘}{q}\left(\Upsilon \right).$$

Here$\;G_{0}^{\theta },\;\Delta G_{S0}^{\theta }\left(\theta ,c\right)\;\mathrm{and}\;A^{S0}\left(\theta ,c\right)$ are functions of temperature θ and concentration c, the order parameter, Υ changed from 0 to 1 from liquid to solid side,$\;G_{0}^{\theta }$ is the free energy of the liquid phase, $f^{L}\left(c_{L}\right)$, and $\Delta G_{S0}^{\theta }\left(\theta ,c\right)=f^{S}\left(c_{S}\right)-f^{L}\left(c_{L}\right)$ is the free energy difference between solid and liquid. $f^{L}\left(c_{L}\right)\;\mathrm{and}\;f^{S}\left(c_{S}\right)\;$ are the free energy densities of liquid and solid as functions of composition, and $A^{S0}\left(\theta ,c\right)$ is the height of the double-well potential.$\;q\left(\Upsilon ,a\right)\;\mathrm{and}\;\overset{˘}{q}\left(\Upsilon \right)$ are connecting functions,
(4)$$q\left(\Upsilon ,a\right)=a\Upsilon^{2}-2\left(a-2\right)\Upsilon^{3}+\left(a-3\right)\Upsilon^{4},$$
(5)$$\overset{˘}{q}\left(\Upsilon \right)=\Upsilon^{2}{\left(1-\Upsilon \right)}^{2}.$$

Here, *a* is a material free parameter. The mole fraction *c* is expressed as,
(6)$$c=\left(c_{S}^{B}-c_{L}^{B}\right)q\left(\Upsilon ,a\right)+c_{L}^{B},$$ where, $c_{S}^{B}\;\;\mathrm{and}\;c_{L}^{B}\;$ are the compositions of bulk solid and liquid phases, respectively. From Equations (2) and (3), local free energy is
(7)$$\psi^{l}=\psi^{\theta }+{\overset{˘}{\psi }}^{\theta }=f\left(c,\Upsilon ,\theta \right)=G_{0}^{\theta }+\Delta G_{S0}^{\theta }\left(\theta ,c\right)q\left(\Upsilon ,a\right)+A^{S0}\left(\theta ,c\right)\overset{˘}{q}\left(\Upsilon \right).$$

We consider the first derivative of solid and liquid free energies with respect to their concentration to be equal, i.e.,
(8)$$f_{c_{S}}^{S}\left[c_{S}\left(x,t\right)\right]=f_{c_{L}}^{L}\left[c_{L}\left(x,t\right)\right].$$

Here, subscripts indicate derivatives. So, $f_{c_{S}}^{S}\left[c_{S}\left(x,t\right)\right]=df^{S}/dc_{S}\;\mathrm{and}\;f_{c_{L}}^{L}\left[c_{L}\left(x,t\right)\right]=df^{L}/dc_{L}.\;$Differentiating Equation (8) with respect to $c_{S}$ and rearranging,
(9)$$\left(\frac{\partial c_{L}}{\partial c_{S}}\right)=\frac{f_{cc}^{S}\left(c_{S}\right)}{f_{cc}^{L}\left(c_{L}\right)},$$ where we used the notations of $f_{cc}^{L}\left(c_{L}\right)=d^{2}f^{L}/dc_{L}^{2}$ and $f_{cc}^{S}\left(c_{L}\right)=d^{2}f^{S}/dc_{S}^{2}$. Differentiating Equation (6) with respect to $c_{L}$, composition of liquid at the interface,
(10)$$\frac{\partial c}{\partial c_{L}}=q\left(\Upsilon ,a\right)\left(\frac{\partial c_{S}}{\partial c_{L}}-1\right)+1.$$

Putting values from Equation (9),
(11)$$\frac{\partial c}{\partial c_{L}}=q\left(\Upsilon ,a\right)\left(\frac{f_{cc}^{L}\left(c_{L}\right)}{f_{cc}^{S}\left(c_{S}\right)}-1\right)+1.$$

Rearranging Equation (11),
(12)$$\frac{\partial c_{L}}{\partial c}=\frac{f_{cc}^{S}\left(c_{S}\right)}{\left[1-q\left(\Upsilon ,a\right)\right]f_{cc}^{S}\left(c_{S}\right)+q\left(\Upsilon ,a\right)f_{cc}^{L}\left(c_{L}\right)}.$$

Similarly, for the solid side,
(13)$$\frac{\partial c_{S}}{\partial c}=\frac{f_{cc}^{L}\left(c_{L}\right)}{\left[1-q\left(\Upsilon ,a\right)\right]f_{cc}^{S}\left(c_{S}\right)+q\left(\Upsilon ,a\right)f_{cc}^{L}\left(c_{L}\right)}.$$

Here, $c_{S}$ is the composition of solid at the interface. Differentiating Equation (6) with respect to Υ,
(14)$$\frac{\partial c}{\partial \Upsilon }=\left(c_{S}^{B}-c_{L}^{B}\right)q^{\prime }\left(\Upsilon ,a\right).$$

Now applying chain rule and Putting value from Equations (12) and (14),
(15)$$\frac{\partial c_{L}}{\partial \Upsilon }=\frac{\left(c_{S}^{B}-c_{L}^{B}\right)q^{\prime }\left(\Upsilon ,a\right)f_{cc}^{S}\left(c_{S}\right)}{\left[1-q\left(\Upsilon ,a\right)\right]f_{cc}^{S}\left(c_{S}\right)+q\left(\Upsilon ,a\right)f_{cc}^{L}\left(c_{L}\right)},$$

Similarly, for the solid side,
(16)$$\frac{\partial c_{S}}{\partial \Upsilon }=\frac{\left(c_{S}^{B}-c_{L}^{B}\right)q^{\prime }\left(\Upsilon ,a\right)f_{cc}^{L}\left(c_{L}\right)}{\left[1-q\left(\Upsilon ,a\right)\right]f_{cc}^{S}\left(c_{S}\right)+q\left(\Upsilon ,a\right)f_{cc}^{L}\left(c_{L}\right)}.$$

The local free energy is
(17)$$\psi^{l}=G_{0}^{\theta }+\Delta G_{S0}^{\theta }\left(\theta ,c\right)q\left(\Upsilon ,a\right)+A^{S0}\left(\theta ,c\right)\overset{˘}{q}\left(\Upsilon \right).$$

Differentiating with respect to Υ,
(18)$$\psi^{l}{}_{\Upsilon }=\frac{\partial }{\partial \Upsilon }\left(G_{0}^{\theta }\right)+q^{\prime }\left(\Upsilon ,a\right)\Delta G_{S0}^{\theta }\left(\theta ,c\right)+q\left(\Upsilon ,a\right)\frac{\partial }{\partial \Upsilon }\left[\Delta G_{S0}^{\theta }\left(\theta ,c\right)\right]+A^{S0}\left(\theta ,c\right){\overset{˘}{q}}^{\prime }\left(\Upsilon \right)\;+\overset{˘}{q}\left(\Upsilon \right)\frac{\partial A^{S0}\left(\theta ,c\right)}{\partial \Upsilon }.$$

Substituting from Equations (15) and (16),
$$\frac{\partial }{\partial \Upsilon }\left[\Delta G_{S0}^{\theta }\left(\theta ,c\right)\right]=\frac{\partial }{\partial \Upsilon }\left[f^{S}\left(c_{S}\right)-f^{L}\left(c_{L}\right)\right]=\frac{\partial }{\partial \Upsilon }\left[f^{S}\left(c_{S}\right)\right]-\frac{\partial }{\partial \Upsilon }\left[f^{L}\left(c_{L}\right)\right]\;\;\;\;\;=\left(\frac{\partial f^{S}\left(c_{S}\right)}{\partial c_{S}}\right)\frac{\partial c_{S}}{\partial \Upsilon }-\left(\frac{\partial f^{L}\left(c_{L}\right)}{\partial c_{L}}\right)\frac{\partial c_{L}}{\partial \Upsilon },$$
$$\Rightarrow \frac{\partial }{\partial \Upsilon }\left[\Delta G_{S0}^{\theta }\left(\theta ,c\right)\right]=\frac{q^{\prime }\left(\Upsilon ,a\right)\frac{\partial f^{L}\left(c_{L}\right)}{\partial c_{L}}\left(c_{S}^{B}-c_{L}^{B}\right)\left[f_{cc}^{L}\left(c_{L}\right)-f_{cc}^{S}\left(c_{S}\right)\right]}{\left[1-q\left(\Upsilon ,a\right)\right]f_{cc}^{S}\left(c_{S}\right)+q\left(\Upsilon ,a\right)f_{cc}^{L}\left(c_{L}\right)}.$$

For $\frac{\partial }{\partial \Upsilon }\left(G_{0}^{\theta }(c_{L)}\right)$ we have,
$$\frac{\partial }{\partial \Upsilon }\left(G_{0}^{\theta }\left(c_{L}\right)\right)=\frac{\partial }{\partial \Upsilon }\left(f^{L}\left(c_{L}\right)\right)=\left(\frac{\partial f^{L}\left(c_{L}\right)}{\partial c_{L}}\right)\frac{\partial c_{L}}{\partial \Upsilon }\;\;\;\;\;\;\;\;\;\;=\frac{\partial f^{L}\left(c_{L}\right)}{\partial c_{L}}\frac{\left(c_{S}^{B}-c_{L}^{B}\right)q^{\prime }\left(\Upsilon ,a\right)f_{cc}^{S}\left(c_{S}\right)}{\left[1-q\left(\Upsilon ,a\right)\right]f_{cc}^{S}\left(c_{S}\right)+q\left(\Upsilon ,a\right)f_{cc}^{L}\left(c_{L}\right)}.$$

Now putting the values of $\frac{\partial }{\partial \Upsilon }\left[\Delta G_{S0}^{\theta }\left(\theta ,c\right)\right]\;\mathrm{and}\;\frac{\partial }{\partial \Upsilon }\left(G_{0}^{\theta }\right)\;$in Equation (18),
(19)$$\begin{array}{c} \psi_{Y}^{l}=q^{\prime }\left(Y,a\right)\Delta G_{S0}^{\theta }\left(\theta ,c\right)+\frac{\partial f^{L}\left(c_{L}\right)}{\partial c_{L}}\frac{\left(c_{S}^{B}-c_{L}^{B}\right)q^{\prime }\left(Y,a\right)f_{cc}^{S}\left(c_{S}\right)}{\left[1-q\left(Y,a\right)\right]f_{cc}^{S}\left(c_{S}\right)+q\left(Y,a\right)f_{cc}^{L}\left(c_{L}\right)}\\ \;\;\;\;\;+q\left(Y,a\right)\left[\frac{q^{\prime }\left(Y,a\right)\frac{\partial f^{L}\left(c_{L}\right)}{\partial c_{L}}\left(c_{S}^{B}-c_{L}^{B}\right)\left[f_{cc}^{S}\left(c_{S}\right)-f_{cc}^{L}\left(c_{L}\right)\right]}{\left[1-q\left(Y,a\right)\right]f_{cc}^{S}\left(c_{S}\right)+q\left(Y,a\right)f_{cc}^{L}\left(c_{L}\right)}\right]\\ \;\;\;\;\;+A^{S0}\left(\theta ,c\right){\overset{˘}{q}}^{\prime }\left(Y\right)+\overset{˘}{q}\left(Y\right)\frac{\partial A^{S0}\left(\theta ,c\right)}{\partial c}\frac{\partial c}{\partial Y}\\ \Rightarrow \psi^{l}{}_{\Upsilon }=q^{\prime }\left(\Upsilon ,a\right)\left[\Delta G_{S0}^{\theta }\left(\theta ,c\right)+\frac{\partial f^{L}\left(c_{L}\right)}{\partial c_{L}}\left(c_{S}^{B}-c_{L}^{B}\right)\right]+A^{S0}\left(\theta ,c\right){\overset{˘}{q}}^{\prime }\left(\Upsilon \right)\\ +\overset{˘}{q}\left(\Upsilon \right)\frac{\partial A^{S0}\left(\theta ,c\right)}{\partial c}\left(c_{S}^{B}-c_{L}^{B}\right)q^{\prime }\left(\Upsilon ,a\right) \end{array}$$

Assuming a second-degree gradient energy term, which is the lowest degree potential function granting a linear relation between the thermodynamic driving force and ∇Υ, we have [36],
(20)$$\psi^{\nabla }=0.5\;\left(\beta^{S0}\nabla \Upsilon^{2}\right)$$

Here, $\beta^{S0\;}$is the gradient energy coefficient. Now, differentiating Equation (20) with respect to Υ,
(21)$$\psi^{\nabla }{}_{\Upsilon }=\nabla .\left[\beta^{S0}\nabla \Upsilon \right].$$

Substituting the value of free energy and gradient energy in Equation (1),
(22)$$\psi =G_{0}^{\theta }+\Delta G_{S0}^{\theta }\left(\theta ,c\right)q\left(\Upsilon ,a\right)+A^{S0}\left(\theta ,c\right)\overset{˘}{q}\left(\Upsilon \right)+0.5\;\left(\beta^{S0}\nabla \Upsilon^{2}\right),$$ and differentiating with respect to Υ,
(23)$$\psi_{\Upsilon }=q^{\prime }\left(\Upsilon ,a\right)\left[\Delta G_{S0}^{\theta }\left(\theta ,c\right)+\frac{\partial f^{L}\left(c_{L}\right)}{\partial c_{L}}\left(c_{S}^{B}-c_{L}^{B}\right)\right]+A^{S0}\left(\theta ,c\right){\overset{˘}{q}}^{\prime }\left(\Upsilon \right)\;\;\;\;\;\;\;\;+\overset{˘}{q}\left(\Upsilon \right)\frac{\partial A^{S0}\left(\theta ,c\right)}{\partial c}\left(c_{S}^{B}-c_{L}^{B}\right){\overset{˘}{q}}^{\prime }\left(\Upsilon \right)+\nabla .\left[\beta^{S0}\nabla \Upsilon \right].$$

The GL equation becomes
(24)$$\frac{1}{L_{\Upsilon }}\frac{\partial \Upsilon }{\partial t}=-\frac{\partial \psi^{l}}{\partial \Upsilon }+\nabla .\left[\beta^{S0}\nabla \Upsilon \right].$$

Here, $L_{\Upsilon }$ is the kinetic coefficient. For simplicity, we assume $A^{S0}\left(\theta ,c\right)$ to be independent of *c*, i.e., $\partial A^{S0}\left(\theta ,c\right)/\partial c=0$. Using Equation (19), we have
(25)$$\frac{1}{L_{\Upsilon }}\frac{\partial \Upsilon }{\partial t}=-\left\{q^{\prime }\left(\Upsilon ,a\right)\left[\Delta G_{S0}^{\theta }\left(\theta ,c\right)+\frac{\partial f^{L}\left(c_{L}\right)}{\partial c_{L}}\left(c_{S}^{B}-c_{L}^{B}\right)\right]+A^{S0}\left(\theta ,c\right){\overset{˘}{q}}^{\prime }\left(\Upsilon \right)\right\}\;+\nabla .\left[\beta^{S0}\nabla \Upsilon \right].$$

Differentiating Equation (17) with respect to *c*,
(26)$$\psi^{l}{}_{c}=q\left(\Upsilon ,a\right)\left(\frac{d}{dc}\left[\Delta G_{S0}^{\theta }\left(\theta ,c\right)\right]\right)+\frac{\partial }{\partial c}\left(G_{0}^{\theta }\right)+\overset{˘}{q}\left(\Upsilon \right)\frac{dA^{S0}\left(\theta ,c\right)}{dc}\;=q\left(\Upsilon ,a\right)\left[\frac{d}{dc}\left[f^{S}\left(c_{S}\right)\right]-\frac{d}{dc}\left[f^{L}\left(c_{L}\right)\right]\right]+\frac{d}{dc}\left[f^{L}\left(c_{L}\right)\right]+\overset{˘}{q}\left(\Upsilon \right)\frac{dA^{S0}\left(\theta ,c\right)}{dc}\;=q\left(\Upsilon ,a\right)\left[\left(\frac{\partial f^{S}\left(c_{S}\right)}{\partial c_{L}}\right)\frac{\partial c_{S}}{\partial c}-\left(\frac{\partial f^{L}\left(c_{L}\right)}{\partial c_{L}}\right)\frac{\partial c_{L}}{\partial c}\right]+\left(\frac{\partial f^{L}\left(c_{L}\right)}{\partial c_{L}}\right)\frac{\partial c_{L}}{\partial c}+\overset{˘}{q}\left(\Upsilon \right)\frac{dA^{S0}\left(\theta ,c\right)}{dc}\;=\left(\frac{\partial f^{L}\left(c_{L}\right)}{\partial c_{L}}\right)\left[q\left(\Upsilon ,a\right)\left(\frac{\partial c_{S}}{\partial c}-\frac{\partial c_{L}}{\partial c}\right)+\frac{\partial c_{L}}{\partial c}\right]+\overset{˘}{q}\left(\Upsilon \right)\frac{dA^{S0}\left(\theta ,c\right)}{dc}.$$

Substituting values from Equations (12) and (13),
(27)$$\psi^{l}{}_{c}=\frac{\partial f^{L}\left(c_{L}\right)}{\partial c_{L}}+\overset{˘}{q}\left(\Upsilon \right)\frac{dA^{S0}\left(\theta ,c\right)}{dc}.$$

Differentiating Equation (26) with respect to c considering, $\frac{\partial }{\partial c}\frac{\partial f^{L}\left(c_{L}\right)}{\partial c_{L}}=\frac{\partial c_{L}}{\partial c}\frac{\partial^{2}f^{L}\left(c_{L}\right)}{\partial c_{L}{}^{2}}$
(28)$$\psi^{l}{}_{cc}=\frac{f_{cc}^{L}\left(c_{L}\right)f_{cc}^{S}\left(c_{S}\right)}{\left[1-q\left(\Upsilon ,a\right)\right]f_{cc}^{S}\left(c_{S}\right)+q\left(\Upsilon ,a\right)f_{cc}^{L}\left(c_{L}\right)}+\overset{˘}{q}\left(\Upsilon \right)\frac{d^{2}A^{S0}\left(\theta ,c\right)}{dc^{2}}.$$

On the other hand, differentiating $\psi^{l}{}_{c}\;\;$with respect to  Υ,
(29)$$\begin{array}{c} \psi^{l}{}_{c\Upsilon }=\frac{d}{d\Upsilon }\left[\frac{df^{L}\left(c_{L}\right)}{dc_{L}}\right]+\frac{dA^{S0}\left(\theta ,c\right)}{dc}{\overset{˘}{q}}^{\prime }\left(\Upsilon \right)\\ =\frac{d^{2}f^{L}\left(c_{L}\right)}{dc_{L}{}^{2}}\frac{\partial c_{L}}{\partial \Upsilon }+\frac{dA^{S0}\left(\theta ,c\right)}{dc}{\overset{˘}{q}}^{\prime }\left(\Upsilon \right)\\ =\frac{\left(c_{S}^{B}-c_{L}^{B}\right)q^{\prime }\left(\Upsilon ,a\right)f_{cc}^{S}\left(c_{S}\right)f_{cc}^{L}\left(c_{L}\right)}{\left[1-q\left(\Upsilon ,a\right)\right]f_{cc}^{S}\left(c_{S}\right)+q\left(\Upsilon ,a\right)f_{cc}^{L}\left(c_{L}\right)}+\frac{dA^{S0}\left(\theta ,c\right)}{dc}{\overset{˘}{q}}^{\prime }\left(\Upsilon \right). \end{array}$$

The Cahn-Hilliard equation, governing the evolution of conserved variables,
(30)$$c_{t}=\nabla .\frac{D\left(\Upsilon \right)}{\psi^{l}{}_{cc}}\nabla \psi^{l}{}_{c}.$$

Here, D(Υ) is the diffusivity. Substituting the value of $\psi^{l}{}_{c}\;$from Equation (26),
(31)$$\frac{\partial c}{\partial t}=\nabla \left[\frac{D\left(\Upsilon \right)}{\psi^{l}{}_{cc}}\nabla \left\{\frac{df^{L}\left(c_{L}\right)}{dc_{L}}+\overset{˘}{q}\left(\Upsilon \right)\frac{dA^{S0}\left(\theta ,c\right)}{dc}\right\}\right].$$

Equations (25) and (31) are the basic equations governing the transformation of binary alloys. We will consider simplified cases of technological importance in the next four subsections, i.e., Section 2.1, Section 2.2, Section 2.3, Section 2.4 and Section 2.5, which are included for completeness and can be omitted if numerical simulations of the model are of interest.

### 2.1. Dilute Solution Approximation

The dilute solution limit is frequently applicable to engineering problems and to study the fundamental physical mechanisms governing the phase transformation. Considering a binary alloy of *A* and *B*, the chemical potential of *A* and *B* can be approximated as [37],
(32)$$\begin{array}{c} \mu_{A}^{L}=\mu_{A}^{oL}+R\theta ln\left(1-c\right);\;\;\mu_{A}^{S}=\mu_{A}^{oS}+R\theta ln\left(1-c\right);\;\\ \mu_{B}^{L}=\mu_{B}^{oL}+R\theta ln\left(\gamma^{L}c\right);\;\;\mu_{B}^{S}=\mu_{B}^{oS}+R\theta ln\left(\gamma^{S}c\right). \end{array}$$

Here, *R* is the gas constant, and θ is the temperature of the isothermal system. $\gamma^{S}\;\mathrm{and}\;\gamma^{L}$ are the activity coefficients of solid and liquid phases, respectively. They are a measure of how much the thermodynamic characteristics of that mixture deviate from those of the ideal mixture. At equilibrium conditions, $\mu_{A}^{L}=\mu_{A}^{S}\;\mathrm{and}\;\mu_{B}^{L}=\mu_{B}^{S}$. Assuming liquid phase as a standard state,
(33)$$\mu_{A}^{oL}=0;\mu_{B}^{oL}=0,$$ and applying this relation of the thermochemical potential at equilibrium concentration, we can rewrite Equation (32) as,
(34)$$\begin{array}{c} \mu_{A}^{L}=\mu_{A}^{S}\Rightarrow \mu_{A}^{oL}+R\theta ln\left(1-c_{L}^{e}\right)=\mu_{A}^{oS}+R\theta ln\left(1-c_{S}^{e}\right)\\ \Rightarrow \mu_{A}^{oS}=R\theta ln\left(\frac{1-c_{L}^{e}}{1-c_{S}^{e}}\right),\\ \mu_{B}^{L}=\mu_{B}^{S}\Rightarrow \mu_{B}^{oL}+R\theta ln\left(\gamma^{L}c_{L}^{e}\right)=\mu_{B}^{oS}+R\theta ln\left(\gamma^{S}c_{S}^{e}\;\right)\\ \Rightarrow \mu_{B}^{oS}=R\theta ln\left(\frac{c_{L}^{e}}{c_{S}^{e}}\right)+R\theta ln\left(\frac{\gamma^{L}}{\gamma^{S}}\right). \end{array}$$

Here, $c_{L}^{e}\;\mathrm{and}\;c_{S}^{e}$, are the equilibrium concentration of the liquid side and solid side, respectively. We can set $\;\gamma^{L}=\gamma^{S}=1$ as these values do not affect the equilibrium state and the driving force for the transformation. These relations are used to derive the free energy density of solid and liquid. We use the following form of the free energy density for the liquid phase [37],
(35)$$\begin{array}{c} f^{L}\left(c_{L}\right)=\frac{\left[\left(1-c_{L}\right)\mu_{A}^{oL}+c_{L}\mu_{B}^{oL}+R\theta \left\{c_{L}ln\left(c_{L}\right)+\left(1-c_{L}\right)ln\left(1-c_{L}\right)\right\}\right]}{V_{m}},\\ \Rightarrow f^{L}\left(c_{L}\right)=\frac{R\theta }{V_{m}}\left[c_{L}lnc_{L}+\left(1-c_{L}\right)ln\left(1-c_{L}\right)\right]. \end{array}$$

Here, $V_{m}$ is the molar volume. Differentiating Equation (35) with respect to $c_{L},$
(36)$$f_{c_{L}}^{L}\left(c_{L}\right)=\frac{R\theta }{V_{m}}ln\left(\frac{c_{L}}{1-c_{L}}\right).$$

Again, differentiating Equation (36) with respect to $c_{L},$
(37)$$f_{cc}^{L}\left(c_{L}\right)=\frac{R\theta }{V_{m}}\frac{1}{\left(1-c_{L}\right)c_{L}}.$$

The free energy density of the solid phase is [37],
(38)$$\begin{array}{c} f^{S}\left(c_{S}\right)=\frac{\left[\left(1-c_{S}\right)\mu_{A}^{oS}+c_{S}\mu_{B}^{oS}+R\theta \left\{c_{S}lnc_{S}+\left(1-c_{S}\right)ln\left(1-c_{S}\right)\right\}\right]}{V_{m}},\\ \Rightarrow f^{S}\left(c_{S}\right)=\frac{R\theta }{V_{m}}\left[\left(1-c_{S}\right)ln\left(\frac{1-c_{L}^{e}}{1-c_{S}^{e}}\right)+c_{S}ln\left(\frac{c_{L}^{e}}{c_{S}^{e}}\right)+c_{S}lnc_{S}+\left(1-c_{S}\right)ln\left(1-c_{S}\right)\right]. \end{array}$$

Differentiating Equation (38) with respect to $c_{S},$
(39)$$f_{c_{S}}^{S}\left(c_{S}\right)=\frac{R\theta }{V_{m}}ln\left(\frac{c_{S}}{1-c_{S}}\;\frac{1-c_{L}^{e}}{1-c_{S}^{e}}\;\frac{c_{L}^{e}}{c_{S}^{e}}\right).$$

Again differentiating Equation (40) with respect to $c_{S},$
(40)$$f_{cc}^{S}\left(c_{S}\right)=\frac{R\theta }{V_{m}}\frac{1}{\left(1-c_{S}\right)c_{S}}ln\left(\;\frac{1-c_{L}^{e}}{1-c_{S}^{e}}\;\frac{c_{L}^{e}}{c_{S}^{e}}\right).$$

Putting the value from Equations (36) and (39) in Equation (8) and reorganizing,
(41)$$\frac{c_{S}^{e}c_{L}}{c_{L}^{e}c_{S}}=\frac{\left(1-c_{S}^{e}\right)\left(1-c_{L}\right)}{\left(1-c_{L}^{e}\right)\left(1-c_{S}\right)}.$$

Now,
(42)$$G\left(c_{S},\;c_{L}\right)\equiv \;\Delta G_{S0}^{\theta }\left(\theta ,c\right)+\frac{df^{L}\left(c_{L}\right)}{dc_{L}}\left(c_{S}^{B}-c_{L}^{B}\right)\;\;\;\;\;\;\;\;\;=f^{L}\left(c_{L}\right)-f^{S}\left(c_{S}\right)-\left(c_{L}-c_{S}\right)f_{c_{L}}^{L}\left(c_{L}\right).$$

Substituting the value from Equations (35), (36) and (38),
(43)$$G\left(c_{S},\;c_{L}\right)=\frac{R\theta }{V_{m}}ln\frac{\left(1-c_{S}^{e}\right)\left(1-c_{L}^{B}\right)}{\left(1-c_{L}^{e}\right)\left(1-c_{S}^{B}\right)}.$$

Using Taylor’s expansion and neglecting the higher-order terms,
(44)$$G\left(c_{S},\;c_{L}\right)=\frac{R\theta }{V_{m}}\left[\left(c_{L}^{e}-c_{S}^{e}\right)-\left(c_{L}-c_{S}\right)\right].$$

At limit where all compositions go to zero Equation (41) can be approximated as
(45)$$\frac{c_{S}}{c_{L}}=\frac{c_{S}^{e}}{c_{L}^{e}}=k^{e},$$ where $k^{e}$ is the equilibrium partition coefficient. Substituting$\;c_{S}=k^{e}c_{L}\;\mathrm{and}\;c_{S}^{e}=k^{e}c_{L}^{e}$ in Equation (44) we can derive,
(46)$$G\left(c_{S},\;c_{L}\right)=\frac{R\theta }{V_{m}}\left[\left(c_{L}^{e}-k^{e}c_{L}^{e}\right)-\left(c_{L}-k^{e}c_{L}\right)\right]=\frac{R\theta }{V_{m}}\frac{1-k^{e}}{m^{e}}\left(m^{e}c_{L}^{e}-m^{e}c_{L}\right),$$

Here, $m^{e}$ is the liquids slope in the phase diagram. For the dilute solution [38],
(47)θ=θm−mecL(1+(ke−k)+kln(kke)1−ke)−VmRθαme1−keVn.

Here, $V_{n}$ is the interface velocity, and $k=c_{S}/c_{L}$. For equilibrium condition, $k^{e}=k$, and interface velocity, $V_{n}=0.\;\theta_{m}$ is the melting temperature of the pure solvent. So, Equation (47) reduces to,
(48)$$\theta =\theta_{m}-m^{e}c_{L}^{e}.$$

From Equation (46),
(49)$$G\left(c_{S},\;c_{L}\right)=\frac{R\theta }{V_{m}}\frac{1-k^{e}}{m^{e}}\left(\theta_{m}-\theta -m^{e}c_{L}\right).$$

Now from Equation (25),
(50)$$\frac{1}{L_{\Upsilon }}\frac{\partial \Upsilon }{\partial t}=-\left\{q^{\prime }\left(\Upsilon ,a\right)G\left(c_{S},\;c_{L}\right)+A^{S0}\left(\theta ,c\right){\overset{˘}{q}}^{\prime }\left(\Upsilon \right)\right\}+\overset{˘}{q}\left(\Upsilon \right)\frac{\partial A^{S0}\left(\theta ,c\right)}{\partial \Upsilon }+\nabla .\left[\beta^{S0}\nabla \Upsilon \right].$$

Substituting the value from Equation (49),
(51)$$\frac{1}{L_{\Upsilon }}\frac{\partial \Upsilon }{\partial t}=-\left\{q^{\prime }\left(\Upsilon ,a\right)\frac{R\theta }{V_{m}}\frac{1-k^{e}}{m^{e}}\left(\theta_{m}-\theta -m^{e}c_{L}\right)+A^{S0}\left(\theta ,c\right){\overset{˘}{q}}^{\prime }\left(\Upsilon \right)\right\}+\nabla .\left[\beta^{S0}\nabla \Upsilon \right].$$

For a dilute solution, the height of double-well potential is constant. Putting the value of Equations (37) and (40), in Equation (28),
(52)$$\psi^{l}{}_{cc}=\frac{R\theta }{V_{m}}\frac{1}{\left(1-q\left(\Upsilon ,a\right)\right)\left(1-c_{L}\right)c_{L}+q\left(\Upsilon ,a\right)\left(1-c_{S}\right)c_{S}}.$$

So,
(53)$$H\left(\Upsilon ,c_{S},c_{L}\right)\equiv \frac{R\theta }{V_{m}\psi^{l}{}_{cc}}=\left(1-q\left(\Upsilon ,a\right)\right)\left(1-c_{L}\right)c_{L}+q\left(\Upsilon ,a\right)\left(1-c_{S}\right)c_{S}.$$

Now putting the value of Equations (36) and (53) in Equation (31),
(54)$$\frac{\partial c}{\partial t}=\nabla \left[D\left(\Upsilon \right)H\left(\Upsilon ,c_{S},c_{L}\right)\nabla ln\left(\frac{c_{L}}{1-c_{L}}\right)\right].$$

In summary, we derived the kinetic GL equations for the dilute solution approximation, i.e., Equations (51) and (54), respectively.

### 2.2. Analytical Solution of Ginzburg-Landau Equation

From Equations (4), (5) and (7) the local free energy is,
(55)$$\psi^{l}=G_{0}^{\theta }+\Delta G_{S0}^{\theta }\left(\theta ,c\right)\left[a\Upsilon^{2}-2\left(a-2\right)\Upsilon^{3}+\left(a-3\right)\Upsilon^{4}\right]+A^{S0}\left(\theta ,c\right)\left[\Upsilon^{2}{\left(1-\Upsilon \right)}^{2}\right].$$

Reorganizing Equation (55), assuming $\;a=0\;\mathrm{and}\;G_{0}^{\theta }=0,$ we have
(56)$$\psi^{l}=A^{S0}\left(\theta ,c\right)\Upsilon^{2}\left[1-\frac{\left(6-P\right)\Upsilon }{3}+\frac{\left(4-P\right)\Upsilon^{2}}{4}\right],\;$$where,
(57)$$P\left(\theta ,c\right)=\frac{12\Delta G_{S0}^{\theta }\left(\theta ,c\right)}{A^{S0}\left(\theta ,c\right)}.$$

Differentiating Equation (56) with respect to Υ,
(58)$$\frac{\partial \psi^{l}}{\partial \Upsilon }=A^{S0}\left(\theta ,c\right)\Upsilon \left(1-\Upsilon \right)\left[2-\left(4-P\right)\Upsilon \right].$$

Now we calculate the maxima,
(59)$$\begin{array}{c} \frac{\partial^{2}\psi^{l}\left(0\right)}{\partial \Upsilon^{2}}=A^{S0}\left(\theta ,c\right);\;\frac{\partial^{2}\psi^{l}\left(1\right)}{\partial \Upsilon^{2}}=2A^{S0}\left(\theta ,c\right)\left(2-P\right);\\ \Upsilon_{3}=\frac{2}{4-p};\psi_{3}^{l}=\frac{\partial^{2}\psi^{l}\left(\Upsilon_{3}\right)}{\partial \Upsilon^{2}}=\frac{4}{3}\frac{A^{S0}\left(\theta ,c\right)\left(3-P\right)}{{\left(4-P\right)}^{3}}, \end{array}$$ where, $\psi^{l}$ is maximum at $\Upsilon_{3}.$ The 1D time-dependent GL equation is,
(60)$$\frac{\partial \Upsilon }{\partial t}=-\lambda \frac{\partial \psi }{\partial \Upsilon }=-\lambda \left(\frac{\partial \psi^{l}}{\partial \Upsilon }-2\beta^{S0}\frac{\partial^{2}\Upsilon }{\partial x^{2}}\right).$$

Here, λ>0  is the kinetic coefficient. Now we rewrite Equation (60) in dimensionless form. The dimensionless potentials and order parameters are,
(61)$$g=m\psi^{l}=B\xi^{2}-\xi^{3}+\xi^{4},\xi =k\Upsilon ,$$ where,
(62)$$B=\frac{9\left(4-P\right)}{4{\left(6-P\right)}^{2}},k=\frac{3\left(4-P\right)}{4\left(6-P\right)},\;m=\frac{81{\left(4-P\right)}^{3}}{A^{S0}\left(\theta ,c\right){\left(6-P\right)}^{4}}=\frac{k^{2}B}{A^{S0}\left(\theta ,c\right)},$$

Here, *k* can be determined using the condition $\frac{\partial g}{\partial \xi }=0$. Now we define $\;\xi_{1}\;\mathrm{and}\;\xi_{2}$,
(63)$$\begin{array}{c} g=\xi^{2}\left(\xi -\xi_{1}\right)\left(\xi -\xi_{2}\right),\;\\ \xi_{1}=0.5\left(1-\sqrt{1-4B}\right),\;\\ \xi_{2}=0.5\left(1+\sqrt{1-4B}\right). \end{array}$$

Introducing new spatial and time variables,
(64)$$y=\frac{k}{\sqrt{\beta^{S0}m}}x=\sqrt{\frac{A^{S0}\left(\theta ,c\right)}{\beta^{S0}B}}x=\frac{2}{3}\sqrt{\frac{A^{S0}\left(\theta ,c\right)}{\beta^{S0}}}\frac{6-P}{\sqrt{4-P}}x,\;\;\;z=\frac{\lambda k^{2}}{m}t.$$

We obtain the dimensionless form of the GL equation,
(65)$$\frac{\partial \xi }{\partial z}=-\left(\frac{\partial g}{\partial \xi }-2\frac{\partial^{2}\xi }{\partial y^{2}}\right).$$

We only consider time-independent solution so, $\frac{\partial \Upsilon }{\partial z}=0$. The resulting equation is
(66)$$\frac{\partial g}{\partial \xi }=2\frac{\partial^{2}\xi }{\partial y^{2}}.$$

Equation (66) is the equation of motion of material point with a mass equal to 2 in the potential field. An energy integral read,
(67)$$\frac{d\xi }{dy}=\sqrt{g-g_{0}},$$ where, $g_{0}\;\mathrm{is}\;\mathrm{an}\;\mathrm{integral}\;\mathrm{constant}.\;\mathrm{At}\;\mathrm{points}\;\frac{d\xi }{dy}=0,\;\mathrm{at}\;\mathrm{the}\;\mathrm{center}\;\mathrm{of}\;\mathrm{nucleus}\;g=g_{0}$. So,
(68)$$g_{GL}^{*}=g_{GL}-g_{o}=g-g_{0}+{\left(\frac{d\xi }{dy}\right)}^{2}=2\left(g-g_{0}\right).$$

Equation (66) has a periodic solution with *n* diffuse interfaces. The total energy per unit area of *n* diffuse interface is given by,
(69)$$e:=\int_{-l}^{l}g_{GL}^{*}dy=2n\int_{-l}^{l}\sqrt{g-g_{0}}d\xi .$$

Here, $l:=\sqrt{\frac{A^{S0}\left(\theta ,c\right)}{\beta^{S0}B}}L,$ 2*L* is the length of a parallelepiped in the x-direction. The energy *e* is finite even for an infinite slab. Imposing the boundary conditions at the end of the slab,
(70)$$\frac{d\xi \left(-l\right)}{dy}=\frac{d\xi \left(l\right)}{dy}=0,$$
(71)$$g\left(-l\right)=g\left(l\right)=g_{o}.$$

Using Equations (61) and (67),
(72)$$y\left(\xi \right)=\int \frac{d\xi }{\sqrt{\left(B\xi^{2}-\xi^{3}+\xi^{4}-g_{o}\right)}}.$$

Now we consider $g\left(\infty \right)=g\left(-\infty \right)=0,\;\mathrm{we}\;\mathrm{find}\;P=0,\;B=\frac{1}{4},\;\xi_{1}=\xi_{2}=\frac{1}{2}.\;g=\xi^{2}{\left(\xi -\frac{1}{2}\right)}^{2}$. The solution of Equation (15) is [39],
(73)$$\xi \left(y\right)={\left[2+\left(1+e^{-\frac{y-y_{o}}{2}}\right)\right]}^{-1},$$
(74)$$\Upsilon \left(x\right)={\left[1+e^{-\sqrt{\frac{A^{S0}\left(\theta ,c\right)}{\beta^{S0}}}\left(x-x_{o}\right)}\right]}^{-1}.$$

The solution is symmetric around$\;x=x_{o}$. The interface energy is given by [39],
(75)$$E=\left(\frac{4}{3}\right)\sqrt{\beta^{S0}\psi_{3}^{l}};$$

The interface thickness is defined as [39],
(76)$$\Delta =p\sqrt{\beta^{S0}/\psi_{3}^{l}};\;\;2.411\leq p\leq 2.667,$$

The relationships for the interface energy and width obtained here are vital for determining the free parameters of the model reproducing these experimentally measurable quantities. 

Considering the free energy of liquid and solid as,
(77)$$f^{L}\left(c_{L}\right)=y=5{\left(x-7\right)}^{4}+30,$$
(78)$$f^{S}\left(c_{S}\right)=y=2{\left(x-3\right)}^{4}+10,$$

Figure 1 shows the free energy curves of solid and liquid by dotted black and solid red curves, respectively. TL is the common tangent line. Now let us consider $x_{1}\equiv c_{L}\;\mathrm{and}\;x_{2}\equiv c_{S}$. Within the interface region, the composition of free energy density is represented as,
(79)$$\psi =f^{S}\left(c_{S}\right)+\left[f^{L}\left(c_{L}\right)-f^{S}\left(c_{S}\right)\right]q\left(\Upsilon \right)+A^{S0}\left(\theta ,c\right)\overset{˘}{q}\left(\Upsilon \right).$$

Here,
(80)$$q\left(\Upsilon \right)=4\Upsilon^{3}-3\Upsilon^{4};\overset{˘}{q}\left(\Upsilon \right)=\Upsilon^{2}{\left(1-\Upsilon \right)}^{2}\;;\;\Upsilon =\frac{c-c_{S}}{c_{L}-c_{S}}.$$

Total energy, ψ is represented by the curve TE in Figure 1. The height of the double-well potential, $A^{S0}\left(\theta ,c\right)$ is assumed to be constant. As the height goes to zero, the curve passes through the intersecting point of Equations (77) and (78).

### 2.3. Thin Interface Limit

Equations (25) and (31) for the steady-state 1D problem, neglecting diffusivity in solid and assuming $A^{S0}\left(\theta ,c\right)$ to be constant, for the thin interface limit, where the interface thickness is small compared to the diffusive boundary layer, become
(81)$$-\frac{V}{L_{\Upsilon }}\frac{\partial \Upsilon }{\partial x}=-\left\{q^{\prime }\left(\Upsilon ,a\right)\left[\Delta G_{S0}^{\theta }\left(\theta ,c\right)+\frac{df^{L}\left(c_{L}\right)}{dc_{L}}\left(c_{S}^{B}-c_{L}^{B}\right)\right]+A^{S0}\left(\theta ,c\right){\overset{˘}{q}}^{\prime }\left(\Upsilon \right)\right\}\;+\beta^{S0}\frac{d^{2}\Upsilon }{dx^{2}};$$
(82)$$-V\frac{\partial c}{\partial x}=\frac{d}{dx}\left[\frac{D\left(\Upsilon \right)}{\psi^{l}{}_{cc}}\frac{d}{dx}\frac{df^{L}\left(c_{L}\right)}{dc_{L}}\right].$$ where, *V* is the interface velocity. Integrating Equation (82),
(83)$$Vc\left(x\right)+\frac{D\left(\Upsilon \right)}{\psi^{l}{}_{cc}}\frac{d}{dx}\frac{df^{L}\left(c_{L}\right)}{dc_{L}}=A,$$

*A* is an integration constant. On the liquid side, Equation (83) gives,
(84)$$\frac{D\left(\Upsilon \right)}{\psi^{l}{}_{cc}}\frac{d}{dx}\frac{df^{L}\left(c_{L}\right)}{dc_{L}}=A-Vc_{L},\;\Rightarrow \frac{D\left(\Upsilon \right)}{\psi^{l}{}_{cc}}\psi^{l}{}_{cc}\frac{dc_{L}}{dx}=A-Vc_{L,}\;\Rightarrow D_{L}\left(\Upsilon \right)\frac{dc_{L}}{dx}=A-Vc_{L}.$$

Similarly, for the solid side, we can derive that,
(85)$$D_{S}\left(\Upsilon \right)\frac{dc_{S}}{dx}=A-Vc_{S}.$$

Assuming $D_{S}\left(\Upsilon \right)\;$to be negligible,
(86)$$A=Vc_{S},$$

Putting the value of A in Equation (83) and integrating, we get the chemical potential profile [38],
(87)$$f_{c}\left(x\right)=f_{c}^{S}\left(c_{S}\right)-V\int_{-\infty }^{x}\frac{\psi^{l}{}_{cc}}{D\left(\Upsilon \right)}\left[c\left(x\right)-c_{S}\right]dx,$$

Here, $f_{c}^{S}\left(c_{S}\right)\;$ is the integration constant, representing the chemical potential of the solid phase. To calculate the chemical potential of liquid, we assume that the thermodynamic partitioning of concentration at the interface occurs sufficiently over the width of −λ<x<λ. Thus, the chemical potential of liquid is
(88)$$f_{c}^{L}\left(c_{L}\right)=f_{c}^{S}\left(c_{S}\right)-V\int_{-\lambda }^{\lambda }\frac{\psi^{l}{}_{cc}}{D\left(\Upsilon \right)}\left[c\left(x\right)-c_{S}^{i}\right]dx,$$ where $c_{S}^{i}\;\mathrm{is}\;\mathrm{the}\;\mathrm{composition}\;\mathrm{at}\;\mathrm{the}\;\mathrm{solid}\;\mathrm{side}\;\left(x=-\lambda \right)$. The chemical potential’s profile across the interface is given by,
(89)$$f_{c}^{L}\left(c_{L}\right)=f_{c}^{S}\left(c_{S}^{i}\right)-V\int_{-\lambda }^{\lambda }\frac{\psi^{l}{}_{cc}}{D\left(\Upsilon \right)}\left[c\left(x\right)-c_{S}^{i}\right]dx,$$

For equilibrium condition,
(90)$$\Delta G_{S0}^{\theta }\left(\theta ,c\right)+f_{c}^{L}\left(c_{L}\right)\left(c_{S}^{B}-c_{L}^{B}\right)=\Delta G_{S0}^{\theta }\left(\theta ,c^{e}\right)-\left(c_{L}^{e}-c_{S}^{e}\right)f_{c_{L}}^{L}\left(c_{L}\right),$$
(91)$$\psi^{l}{}_{cc}=\psi^{le}{}_{cc},$$

Now multiplying Equation (81) with $\frac{d\Upsilon }{dx}\;$ and integrating from −λ to λ gives,
(92)$$\begin{array}{c} \frac{V}{L_{\Upsilon }}\int_{-\lambda }^{\lambda }{\left(\frac{d\Upsilon }{dx}\right)}^{2}dx=-\beta^{S0}\int_{-\lambda }^{\lambda }\frac{d^{2}\Upsilon }{dx^{2}}\frac{d\Upsilon }{dx}dx\\ \;\;\;\;\;\;\;+\int_{-\lambda }^{\lambda }\left(q^{\prime }\left(\Upsilon ,a\right)\left[\Delta G_{S0}^{\theta }\left(\theta ,c^{e}\right)-\left(c_{L}^{e}-c_{S}^{e}\right)\;f_{c_{L}}^{L}\left(c_{L}\right)\right]\right)\frac{d\Upsilon }{dx}dx\\ \;\;\;\;\;\;\;+\int_{-\lambda }^{\lambda }A^{S0}\left(\theta ,c\right){\overset{˘}{q}}^{\prime }\left(\Upsilon \right)\frac{d\Upsilon }{dx}dx. \end{array}$$

The first term of the right side of Equation (92),
(93)$$-\beta^{S0}\int_{-\lambda }^{\lambda }\frac{d^{2}\Upsilon }{dx^{2}}\frac{d\Upsilon }{dx}dx=\beta^{S0}\int_{0}^{0}\frac{d\Upsilon }{dx}d\left(\frac{d\Upsilon }{dx}\right)=0.$$

The middle term of the right side of Equation (92),
(94)$$\begin{array}{c} \int_{-\lambda }^{\lambda }\left(q^{\prime }\left(\Upsilon ,a\right)\left[\Delta G_{S0}^{\theta }\left(\theta ,c^{e}\right)-\left(c_{L}^{e}-c_{S}^{e}\right)f_{c_{L}}^{L}\left(c_{L}\right)\right]\right)\frac{d\Upsilon }{dx}dx\\ \;\;\;\;\;\;\;\;=\int_{1}^{0}\left[\Delta G_{S0}^{\theta }\left(\theta ,c^{e}\right)-\left(c_{L}^{e}-c_{S}^{e}\right)f_{c}^{S}\left(c_{S}^{i}\right)\right]q^{\prime }\left(\Upsilon ,a\right)d\Upsilon \\ \;\;\;\;\;\;\;\;-\int_{1}^{0}\left(c_{L}^{e}-c_{S}^{e}\right)\left[V\int_{-\lambda }^{\lambda }\frac{\psi^{l}{}_{cc}}{D\left(\Upsilon \right)}\left[c\left(x\right)-c_{S}^{i}\right]dx\right]q^{\prime }\left(\Upsilon ,a\right)d\Upsilon . \end{array}$$
$$\begin{array}{c} \Rightarrow \int_{-\lambda }^{\lambda }\left(q^{\prime }\left(\Upsilon ,a\right)\left[\Delta G_{S0}^{\theta }\left(\theta ,c^{e}\right)-\left(c_{L}^{e}-c_{S}^{e}\right)f_{c_{L}}^{L}\left(c_{L}\right)\right]\right)\frac{d\Upsilon }{dx}dx\\ \;\;\;\;\;\;\;\;=\Delta G_{S0}^{\theta }\left(\theta ,c^{e}\right)-\left(c_{L}^{e}-c_{S}^{e}\right)f_{c}^{S}\left(c_{S}^{i}\right)\\ \;\;\;\;\;\;\;\;-\int_{1}^{0}\left(c_{L}^{e}-c_{S}^{e}\right)\left[V\int_{-\lambda }^{\lambda }\frac{\psi^{l}{}_{\mathrm{cc}}}{D\left(\Upsilon \right)}\left[c\left(x\right)-c_{S}^{i}\right]dx\right]q^{\prime }\left(\Upsilon ,a\right)d\Upsilon , \end{array}$$

The third term of the right side of Equation (92),
(95)$$\int_{-\lambda }^{\lambda }A^{S0}\left(\theta ,c\right){\overset{˘}{q}}^{\prime }\left(\Upsilon \right)\frac{d\Upsilon }{dx}dx=A^{S0}\left(\theta ,c\right)\int_{1}^{0}{\overset{˘}{q}}^{\prime }\left(\Upsilon \right)d\Upsilon =0.$$

Putting these values from Equations (93)–(95) in (92),
(96)$$\frac{V}{L_{\Upsilon }}\int_{-\lambda }^{\lambda }{\left(\frac{d\Upsilon }{dx}\right)}^{2}dx=\Delta G_{S0}^{\theta }\left(\theta ,c^{e}\right)-\left(c_{L}^{e}-c_{S}^{e}\right)f_{c}^{S}\left(c_{S}^{i}\right)\;\;\;\;\;\;\;\;\;\;\;\;\;\;\;\;\;\;-\int_{1}^{0}\left(c_{L}^{e}-c_{S}^{e}\right)\left[V\int_{-\lambda }^{\lambda }\frac{\psi^{l}{}_{cc}}{D\left(\Upsilon \right)}\left[c\left(x\right)-c_{S}^{i}\right]dx\right]q^{\prime }\left(\Upsilon ,a\right)d\Upsilon ,$$

From Equation (14) for equilibrium condition,
(97)$$\frac{dc}{d\Upsilon }=\left(c_{S}-c_{L}\right)q^{\prime }\left(\Upsilon ,a\right)≅-\left(c_{L}^{e}-c_{S}^{e}\right)q^{\prime }\left(\Upsilon ,a\right).$$

So, Equation (96) can be rewritten as,
(98)$$\frac{V}{L_{\Upsilon }}\int_{-\lambda }^{\lambda }{\left(\frac{d\Upsilon }{dx}\right)}^{2}dx=f^{L}\left(c_{L}^{e}\right)-f^{S}\left(c_{S}^{e}\right)-\left(c_{L}^{e}-c_{S}^{e}\right)f_{c}^{S}\left(c_{S}^{i}\right)\;\;\;\;\;\;\;\;\;+\int_{c_{S}}^{c_{L}}V\left[\int_{-\lambda }^{\lambda }\frac{\psi^{l}{}_{cc}}{D\left(\Upsilon \right)}\left[c\left(x\right)-c_{S}^{i}\right]dx\right]dc,$$

From Equation (96) we can show that,
(99)$$\Delta G_{S0}^{\theta }\left(\theta ,c^{e}\right)-\left(c_{L}^{e}-c_{S}^{e}\right)f_{c}^{S}\left(c_{S}^{i}\right)=\alpha V,$$ where,
(100)$$\alpha =\frac{1}{L_{\Upsilon }}\int_{-\lambda }^{\lambda }{\left(\frac{d\Upsilon }{dx}\right)}^{2}dx+\int_{1}^{0}\left(c_{L}^{e}-c_{S}^{e}\right)\left[\int_{-\lambda }^{\lambda }\frac{\psi^{l}{}_{cc}}{D\left(\Upsilon \right)}\left[c\left(x\right)-c_{S}^{i}\right]dx\right]q^{\prime }\left(\Upsilon ,a\right)d\Upsilon .$$

From common tangent relation in equilibrium,
(101)$$f_{c_{L}}^{L}\left(c_{L}^{e}\right)=f_{c_{S}}^{S}\left(c_{S}^{e}\right)=\frac{f^{L}\left(c_{L}^{e}\right)-f^{S}\left(c_{S}^{e}\right)}{c_{L}^{e}-c_{S}^{e}}=\frac{\Delta G_{S0}^{\theta }\left(\theta ,c^{e}\right)}{c_{L}^{e}-c_{S}^{e}}.$$

From Equation (99),
(102)$$\left(c_{L}^{e}-c_{S}^{e}\right)\left[f_{c_{S}}^{S}\left(c_{S}^{e}\right)-f_{c}^{S}\left(c_{S}^{i}\right)\right]=\alpha V.$$

From dilute solution approximation,
(103)$$f_{c_{S}}^{S}\left(c_{S}^{e}\right)-f_{c}^{S}\left(c_{S}^{i}\right)=\frac{R\theta }{v_{m}}\left(1-\frac{c_{S}^{i}}{c_{S}^{e}}\right).$$

From Equations (102) and (103),
$$\begin{array}{c} \left(c_{L}^{e}-c_{S}^{e}\right)\left[\frac{R\theta }{V_{m}}\left(1-\frac{c_{S}^{i}}{c_{S}^{e}}\right)\right]=\alpha V\\ \Rightarrow \left(c_{L}^{e}-c_{S}^{e}\right)\left(1-\frac{c_{S}^{i}}{c_{S}^{e}}\right)=V\frac{V_{m}}{R\theta }\alpha \\ \Rightarrow c_{L}^{e}-c_{S}^{e}-\frac{c_{S}^{i}}{c_{S}^{e}}.c_{L}^{e}+c_{S}^{i}=V\frac{V_{m}}{R\theta }\alpha \\ \Rightarrow c_{L}^{e}\left(1-k^{e}\right)-\frac{c_{S}^{i}}{k^{e}}\left(1-k^{e}\right)=V\frac{V_{m}}{R\theta }\alpha \\ \Rightarrow c_{L}^{e}m^{e}-m^{e}\frac{c_{S}^{i}}{k^{e}}=V\frac{V_{m}}{R\theta }\frac{\alpha m^{e}}{1-k^{e}}\\ \Rightarrow \theta_{m}-\theta -m^{e}\frac{c_{S}^{i}}{k^{e}}=V\frac{V_{m}}{R\theta }\frac{\alpha m^{e}}{1-k^{e}}\\ \Rightarrow \theta =\theta_{m}-m^{e}\frac{c_{S}^{i}}{k^{e}}-V\frac{V_{m}}{R\theta }\frac{\alpha m^{e}}{1-k^{e}} \end{array}$$
(104)$$\theta =\theta_{m}-m^{e}\frac{c_{S}^{i}}{k^{e}}-V\beta ,$$ where$\;\beta =\frac{V_{m}}{R\theta }\frac{\alpha m^{e}}{1-k^{e}}$. Equation (104) is the relationship in the classical sharp interface model between the temperature and composition, proving thermodynamic consistency of the proposed phase-field model.

### 2.4. Solute Trapping

Solute trapping is known as the dependence of jump in concentration through the interface on the interface velocity. The chemical potential depends on the position across the moving interface. The equality of the chemical potential implies that there is no composition gradient across the interface. Solute trapping occurs when the chemical potential varies across the moving interface. We considered a steady-state 1D, Equation (82), dilute solution with constant diffusivity, $D_{i}$, in both interfacial region and liquid phase and negligible diffusivity in the solid. We have,
(105)$$-V\frac{\partial c}{\partial x}=\frac{d}{dx}\left[\frac{D_{i}}{\psi^{l}{}_{cc}}\frac{d}{dx}\frac{df^{L}\left(c_{L}\right)}{dc_{L}}\right].$$

Here, *V* is the interface velocity. Integrating Equation (105),
(106)$$Vc\left(x\right)+\frac{D_{i}}{\psi^{l}{}_{cc}}\frac{d}{dx}\frac{df^{L}\left(c_{L}\right)}{dc_{L}}=A.$$

On the solid side of the interface from Equation (86),
(107)$$A=Vc_{S}^{i}.$$

Putting the value of *A* in Equation (106),
(108)$$\frac{d}{dx}\frac{df^{L}\left(c_{L}\right)}{dc_{L}}=-\frac{V}{D_{i}}\left(c-c_{S}^{i}\right)\psi^{l}{}_{cc}.$$

Now,
(109)$$\frac{d}{dx}\frac{df^{L}\left(c_{L}\right)}{dc_{L}}=\frac{d^{2}f^{L}\left(c_{L}\right)}{dc_{L}{}^{2}}\frac{dc_{L}}{dx}=f_{cc}^{L}\left(c_{L}\right)\frac{dc_{L}}{dx}.$$

Substituting the value from Equation (37),
(110)$$\frac{d}{dx}\frac{df^{L}\left(c_{L}\right)}{dc_{L}}=\frac{RT}{V_{m}}\frac{1}{\left(1-c_{L}\right)c_{L}}\frac{dc_{L}}{dx}.$$

For dilute alloy, $\left(1-c_{L}\right)\to 1\;\mathrm{and}\;\left(1-c_{S}\right)\to 1$. So, from Equation (110),
(111)$$\frac{d}{dx}\frac{df^{L}\left(c_{L}\right)}{dc_{L}}=\frac{RT}{V_{m}}\frac{1}{c_{L}}\frac{dc_{L}}{dx}.$$

Additionally, from Equation (47),
(112)$$\frac{c_{S}\left(x\right)}{c_{L}\left(x\right)}=\frac{c_{S}^{e}}{c_{L}^{e}}=k^{e}.$$

Substituting this relation in Equation (6),
(113)$$c\left(x\right)=[1-\left(1-k^{e}\right)q\left(\Upsilon ,a\right)]c_{L}.$$

From Equations (27), (37) and (40),
(114)$$\psi^{l}{}_{cc}=\frac{RT}{v_{m}}\frac{\frac{1}{\left(1-c_{L}\right)c_{L}}\frac{1}{\left(1-c_{S}\right)c_{S}}}{\left[1-q\left(\Upsilon ,a\right)\right]\frac{1}{\left(1-c_{S}\right)c_{S}}+q\left(\Upsilon ,a\right)\frac{1}{\left(1-c_{L}\right)c_{L}}},\;$$ which reduces to,
(115)$$\psi^{l}{}_{cc}=\frac{RT}{v_{m}[1-\left(1-k^{e}\right)q\left(\Upsilon ,a\right)]c_{L}}.$$

Putting the values from Equations (111) and (115) in Equation (108),
(116)$$\frac{RT}{V_{m}}\frac{1}{c_{L}}\frac{dc_{L}}{dx}=-\frac{V}{D_{i}}\left(c-c_{S}^{i}\right)\frac{RT}{v_{m}[1-\left(1-k^{e}\right)q\left(\Upsilon ,a\right)]c_{L}}.$$

Putting the value of *c* from Equation (113) and simplifying Equation (116),
(117)$$\frac{dc_{L}}{dx}+\frac{V}{D_{i}}c_{L}=\frac{V}{D_{i}}\frac{c_{S}^{i}}{\left[1-\left(1-k^{e}\right)q\left(\Upsilon ,a\right)\right]}.$$

Equation (117) can be rewritten as,
(118)$$y^{\prime }\left(x\right)+ay\left(x\right)=\frac{ab}{1-\left(1-c\right)f\left(x\right)},$$ where, $a=V/D_{i},\;b=c_{S}^{i},\;c=k^{e}\;\mathrm{and}\;q\left(\Upsilon ,a\right)=f\left(x\right)$. The general solution of Equation (118) is
(119)$$y\left(x\right)=ke^{-ax}+e^{-ax}\int_{1}^{x}\frac{abe^{ax^{\prime }}}{1-\left(1-c\right)f\left(x^{\prime }\right)}dx^{\prime }.$$

Putting the values of *a*, *b*, and *c* in Equation (119),
(120)$$c_{L}\left(x\right)=ke^{-\frac{V}{D_{i}}x}+e^{-\frac{V}{D_{i}}x}\int_{1}^{x}\frac{\frac{V}{D_{i}}c_{S}^{i}e^{\frac{V}{D_{i}}x^{\prime }}}{1-\left(1-k^{e}\right)q\left(x^{\prime }\right)}dx^{\prime }.$$

Under the boundary condition $c_{L}=\frac{c_{S}}{k^{e}}=c_{S}^{i}/k^{e}\;\mathrm{at}\;x=-\lambda ,$ we have
(121)$$c_{L}\left(x\right)=\frac{c_{S}^{i}}{k^{e}}e^{-\frac{V}{D_{i}}\left(x+\lambda \right)}+\frac{V}{D_{i}}c_{S}^{i}e^{-\frac{V}{D_{i}}x}\int_{-\lambda }^{x}\frac{e^{\frac{V}{D_{i}}x^{\prime }}}{1-\left(1-k^{e}\right)q\left(x^{\prime }\right)}dx^{\prime }.$$

Substituting in Equation (113),
(122)$$c\left(x\right)=\left[1-\left(1-k^{e}\right)q\left(\Upsilon ,a\right)\right]\left[\frac{c_{S}^{i}}{k^{e}}e^{-\frac{V}{D_{i}}\left(x+\lambda \right)}\;\;\;\;\;\;\;\;\;\;\;\;\;+\frac{V}{D_{i}}c_{S}^{i}e^{-\frac{V}{D_{i}}x}\int_{-\lambda }^{x}\frac{e^{\frac{V}{D_{i}}x^{\prime }}}{1-\left(1-k^{e}\right)q\left(x^{\prime }\right)}dx^{\prime }\right].$$

This equation expresses the equilibrium partition coefficient as a function of interface velocity. Equilibrium partition coefficient *k* is defined as the ratio of composition of the solid side to the liquid side of the interface or composition of the solid side of the interface to the maximum composition across the interface [40]. Equation (122) can be rewritten in dimensionless form as,
(123)$$\tilde{c}\left(x\right)=\left[1-\left(1-k^{e}\right)q\left(\Upsilon ,a\right)\right]\left[\frac{1}{k^{e}}e^{-P\tilde{x}}+Pe^{-P\left(\tilde{x}+\frac{1}{2}\right)}\int_{-0.5}^{x}\frac{e^{P{\tilde{x}}^{\prime }}}{1-\left(1-k^{e}\right)q\left(\Upsilon \right)}d{\tilde{x}}^{\prime }\right],$$ where $\tilde{x}=x/2\lambda \;\mathrm{and}\;\tilde{c}=c/c_{S}^{i}.$ Here, interface Péclet number, *P* = 2*λV*/*D*, controls partition coefficient *k*. We adopted *α* = 2.94 and *k^e^* = 0.8 with which *ϕ* changes from 0.05 to 0.95 at −*λ* < *x* < *λ*.

Figure 2 shows, for small *P* values, the value of $\tilde{c}=c_{\max}/c_{S}^{i}\;$is close to equilibrium. With increasing *P*, the height of the concentration profile and thickness of the diffusive boundary layer decrease around the interface.

Figure 3 shows the variation of partition coefficient, *k*, as a function of interface Péclet number *P*, indicating that the partition coefficient starts from 0.85 and gradually reaches 1 as *P* increases. From the definition, interface thickness can be defined as,
(124)$$k\left(P\right)=\frac{k^{e}+\gamma P}{1+\gamma P},$$ where,
(125)$$\gamma =\frac{8\left(1-k^{e}\right)}{6\alpha \ln\left(\frac{1}{k^{e}}\right)}.$$

Partitionless solidification occurs with complete solute trapping. The interface temperature $\theta \;\mathrm{is}\;\mathrm{below}\;\theta_{o}\;\mathrm{temperature}\;\mathrm{when}\;f^{S}\left(c_{S}\right)\;\mathrm{and}\;f^{L}\left(c_{L}\right)$ become equal. For dilute solution we have,
(126)$$\theta <\theta_{o}=\theta_{m}+c_{\infty }\frac{m^{e}lnk^{e}}{1-k^{e}},$$ where $c_{\infty }\;$ is the bulk composition. During partitionless solidification, the interface velocity is expressed $\;\mathrm{as}\;\left(\theta_{o}-\theta \right)/\beta$ [41]. For dilute solution 1D phase-field equation from Equations (44), (46) and (81),
(127)$$-\frac{V}{L_{\Upsilon }}\frac{\partial \Upsilon }{\partial x}=-\left\{q^{\prime }\left(\Upsilon ,a\right)\frac{R\theta }{V_{m}}\left[\left(c_{L}^{e}-c_{S}^{e}\right)-\left(c_{L}-c_{S}\right)\right]+A^{S0}\left(\theta ,c\right){\overset{˘}{q}}^{\prime }\left(\Upsilon \right)\right\}+\beta^{S0}\frac{d^{2}\Upsilon }{dx^{2}}.$$

Additionally, for partitionless solidification, from Equation (113),
(128)$$c_{L}=\frac{c_{\infty }}{\left[1-\left(1-k^{e}\right)q\left(\Upsilon ,a\right)\right]}.$$

With approximation from Equation (112),
(129)$$-\frac{V}{L_{\Upsilon }}\frac{V_{m}}{RT}\frac{\partial \Upsilon }{\partial x}=-\left\{q^{\prime }\left(\Upsilon ,a\right)c_{L}^{e}\left(1-k^{e}\right)-q^{\prime }\left(\Upsilon ,a\right)\frac{c_{\infty }\left(1-k^{e}\right)}{\left[1-\left(1-k^{e}\right)q\left(\Upsilon ,a\right)\right]}\;+A^{S0}\left(\theta ,c\right){\overset{˘}{q}}^{\prime }\left(\Upsilon \right)\right\}+\beta^{S0}\frac{d^{2}\Upsilon }{dx^{2}}.$$

Now,
(130)$$\frac{d}{d\Upsilon }ln\left[1-\left(1-k^{e}\right)q\left(\Upsilon ,a\right)\right]=-q^{\prime }\left(\Upsilon ,a\right)\frac{\left(1-k^{e}\right)}{\left[1-\left(1-k^{e}\right)q\left(\Upsilon ,a\right)\right]}.$$

Putting the value in Equation (129),
(131)$$-\frac{V}{L_{\Upsilon }}\frac{V_{m}}{R\theta }\frac{\partial \Upsilon }{\partial x}=-\left\{q^{\prime }\left(\Upsilon ,a\right)c_{L}^{e}\left(1-k^{e}\right)+c_{\infty }\frac{d}{d\Upsilon }ln\left[1-\left(1-k^{e}\right)q\left(\Upsilon ,a\right)\right]\;+A^{S0}\left(\theta ,c\right){\overset{˘}{q}}^{\prime }\left(\Upsilon \right)\right\}+\beta^{S0}\frac{d^{2}\Upsilon }{dx^{2}}.$$

Multiplying with $\frac{\partial \Upsilon }{\partial x}\;\mathrm{on}\;\mathrm{both}\;\mathrm{sides}\;\mathrm{and}\;\mathrm{integrate}\;\mathrm{from}\;x=-\infty \;\mathrm{to}\;x=+\infty$,
(132)$$\begin{array}{c} -\frac{V}{L_{\Upsilon }}\frac{V_{m}}{R\theta }\int_{-\infty }^{+\infty }{\left(\frac{\partial \Upsilon }{\partial x}\right)}^{2}dx\\ \;\;\;\;\;\;\;\;=-\int_{-\infty }^{+\infty }q^{\prime }\left(\Upsilon ,a\right)c_{L}^{e}\left(1-k^{e}\right)\frac{\partial \Upsilon }{\partial x}dx\\ \;\;\;\;\;\;\;\;-\int_{-\infty }^{+\infty }c_{\infty }\frac{d}{d\Upsilon }ln\left[1-\left(1-k^{e}\right)q\left(\Upsilon ,a\right)\right]\frac{\partial \Upsilon }{\partial x}dx\\ \;\;\;\;\;\;\;\;-\int_{-\infty }^{+\infty }A^{S0}\left(\theta ,c\right){\overset{˘}{q}}^{\prime }\left(\Upsilon \right)\frac{\partial \Upsilon }{\partial x}dx+\int_{-\infty }^{+\infty }\beta^{S0}\frac{d^{2}\Upsilon }{dx^{2}}\frac{\partial \Upsilon }{\partial x}dx. \end{array}$$

Similarly, from Equations (93) and (95),
(133)$$\int_{-\infty }^{+\infty }A^{S0}\left(\theta ,c\right){\overset{˘}{q}}^{\prime }\left(\Upsilon \right)\frac{\partial \Upsilon }{\partial x}dx=0,$$ and,
(134)$$\int_{-\infty }^{+\infty }\beta^{S0}\frac{d^{2}\Upsilon }{dx^{2}}\frac{\partial \Upsilon }{\partial x}dx=0.$$

Now,
(135)$$\int_{-\infty }^{+\infty }q^{\prime }\left(\Upsilon ,a\right)c_{L}^{e}\left(1-k^{e}\right)\frac{\partial \Upsilon }{\partial x}dx=\int_{0}^{1}q^{\prime }\left(\Upsilon ,a\right)c_{L}^{e}\left(1-k^{e}\right)d\Upsilon =c_{L}^{e}\left(1-k^{e}\right).$$

Additionally,
(136)$$\int_{-\infty }^{+\infty }c_{\infty }\frac{d}{d\Upsilon }ln\left[1-\left(1-k^{e}\right)q\left(\Upsilon ,a\right)\right]\frac{\partial \Upsilon }{\partial x}dx\;\;\;\;\;\;\;\;\;\;\;\;\;\;\;\;=\int_{0}^{1}c_{\infty }\frac{d}{d\Upsilon }ln\left[1-\left(1-k^{e}\right)q\left(\Upsilon ,a\right)\right]d\Upsilon =c_{\infty }lnk^{e}.$$

Putting the values,
(137)$$\frac{V}{L_{\Upsilon }}\frac{V_{m}}{RT}\int_{-\infty }^{+\infty }{\left(\frac{\partial \Upsilon }{\partial x}\right)}^{2}dx=c_{L}^{e}\left(1-k^{e}\right)+c_{\infty }lnk^{e},$$

Therefore, the condition for partitionless solidification is,
(138)$$c_{L}^{e}\left(1-k^{e}\right)+c_{\infty }lnk^{e}>0.$$

### 2.5. Equilibrium and Stability Conditions for the Homogenous Phase

From Equation (19) we get,
(139)$$\psi^{l}{}_{\Upsilon }=q^{\prime }\left(\Upsilon ,a\right)\left[\Delta G_{S0}^{\theta }\left(\theta ,c\right)+\frac{\partial f^{L}\left(c_{L}\right)}{\partial c_{L}}\left(c_{S}^{B}-c_{L}^{B}\right)\right]+A^{S0}\left(\theta ,c\right){\overset{˘}{q}}^{\prime }\left(\Upsilon \right)\;+\overset{˘}{q}\left(\Upsilon \right)\frac{\partial A^{S0}\left(\theta ,c\right)}{\partial c}\left(c_{S}^{B}-c_{L}^{B}\right)q^{\prime }\left(\Upsilon ,a\right).$$

For simplicity, let us assume $A^{S0}\left(\theta ,c\right)\;\mathrm{is}\;\mathrm{constant}.\;\mathrm{So}\;\frac{\partial A^{S0}\left(\theta ,c\right)}{\partial c}=0$.
(140)$$\psi^{l}{}_{\Upsilon }=q^{\prime }\left(\Upsilon ,a\right)\left[\Delta G_{S0}^{\theta }\left(\theta ,c\right)+\frac{\partial f^{L}\left(c_{L}\right)}{\partial c_{L}}\left(c_{S}^{B}-c_{L}^{B}\right)\right]+A^{S0}\left(\theta ,c\right){\overset{˘}{q}}^{\prime }\left(\Upsilon \right),$$

Differentiating Equation (140) with respect to Υ we have,
(141)$$\begin{array}{c} \psi^{l}{}_{\Upsilon \Upsilon }=q^{\prime \prime }\left(\Upsilon ,a\right)\left[\Delta G_{S0}^{\theta }\left(\theta ,c\right)+\frac{\partial f^{L}\left(c_{L}\right)}{\partial c_{L}}\left(c_{S}^{B}-c_{L}^{B}\right)\right]\\ \;\;\;\;\;\;+q^{\prime }\left(\Upsilon ,a\right)\frac{\partial }{\partial \Upsilon }\left[\Delta G_{S0}^{\theta }\left(\theta ,c\right)+\frac{\partial f^{L}\left(c_{L}\right)}{\partial c_{L}}\left(c_{S}^{B}-c_{L}^{B}\right)\right]\\ \;\;\;\;\;\;+A^{S0}\left(\theta ,c\right){\overset{˘}{q}}^{\prime \prime }\left(\Upsilon \right) \end{array},$$

Assuming *a =* 0, we have,
(142)$$q\left(\Upsilon \right)=4\Upsilon^{3}-3\Upsilon^{4},\;q^{\prime }\left(\Upsilon \right)=12\Upsilon^{2}-12\Upsilon^{3},\;q^{\prime \prime }\left(\Upsilon \right)=24\Upsilon -36\Upsilon^{2}.$$

We calculate the values at Υ=0 and Υ=1,
(143)$$q\left(\Upsilon =0\right)=0;\;\;\;\;\;q^{\prime }\left(\Upsilon =0\right)=0;\;\;\;\;\;\;q^{\prime \prime }\left(\Upsilon =0\right)=0,\;q\left(\Upsilon =1\right)=1;\;\;\;\;\;q^{\prime }\left(\Upsilon =1\right)=0;\;\;\;\;\;\;q^{\prime \prime }\left(\Upsilon =1\right)=-12.$$

We also have,
(144)$$\overset{˘}{q}\left(\Upsilon \right)=\Upsilon^{2}{\left(1-\Upsilon \right)}^{2},\;{\overset{˘}{q}}^{\prime }\left(\Upsilon \right)=2\Upsilon -6\Upsilon^{2}+4\Upsilon^{3},\;{\overset{˘}{q}}^{\prime \prime }\left(\Upsilon \right)=2-12\Upsilon +12\Upsilon^{2}.$$

We calculate the values at Υ=0 and Υ=1,
(145)$$\overset{˘}{q}\left(\Upsilon =0\right)=0;\;\;\;\;\;{\overset{˘}{q}}^{\prime }\left(\Upsilon =0\right)=0;\;\;\;\;\;\;\;{\overset{˘}{q}}^{\prime \prime }\left(\Upsilon =0\right)=2,\;\overset{˘}{q}\left(\Upsilon =1\right)=0;\;\;\;\;\;{\overset{˘}{q}}^{\prime }\left(\Upsilon =1\right)=0;\;\;\;\;\;\;\;{\overset{˘}{q}}^{\prime \prime }\left(\Upsilon =1\right)=2.$$

From Equation (141),
(146)$${\frac{\partial^{2}\psi^{l}}{\partial \Upsilon^{2}}|}_{\left(\Upsilon =0\right)}=2A^{S0}\left(\theta ,c\right);$$
(147)$${\frac{\partial^{2}\psi^{l}}{\partial \Upsilon^{2}}|}_{\left(\Upsilon =1\right)}=2A^{S0}\left(\theta ,c\right)-12\left[\Delta G_{S0}^{\theta }\left(\theta ,c\right)+\frac{\partial f^{L}\left(c_{L}\right)}{\partial c_{L}}\left(c_{S}^{B}-c_{L}^{B}\right)\right].$$

Equations (146) and (147) give the value of $\frac{\partial^{2}\psi^{l}}{\partial \Upsilon^{2}}\;\mathrm{at}\;\Upsilon =0\;\mathrm{and}\;\Upsilon =1,$ respectively. At, Υ=0, for M→S phase transformation,
(148)$$\frac{\partial^{2}\psi^{l}}{\partial \Upsilon^{2}}\leq 0,\;A^{S0}\leq 0.$$

At Υ=1, for S→M phase transformation,
(149)$$\begin{array}{c} \frac{\partial^{2}\psi^{l}}{\partial \Upsilon^{2}}\leq 0;\\ A^{S0}-6\left[\Delta G_{S0}^{\theta }\left(\theta ,c\right)+\frac{\partial f^{L}\left(c_{L}\right)}{\partial c_{L}}\left(c_{S}^{B}-c_{L}^{B}\right)\right]\leq 0. \end{array}$$

Now let $\Delta G_{S0}^{\theta }\left(\theta ,c\right)=-\Delta S_{S0}\left(\theta -\theta_{e}^{S0}\right),\mathrm{here}\;\Delta S_{S0}<0$ is the difference in entropy between the solid and liquid phases. $\theta_{e}^{S0}$ is the thermodynamic equilibrium melting temperature of the solid. $A^{S0}=A_{c}^{S0}\left(\theta -\theta_{c}^{S0}\right),\;\mathrm{where}\;\theta_{c}^{S0}$ is the critical temperature where liquid loses its stability.

### 2.6. Numerical Simulation

Let us consider a 1D isothermal system with uniform bulk modulus. The system temperature with undercooling is given. When the system temperature is lower than the solidus, the system can reach a steady state. The system can also reach a steady state when a solute sink exists and sweep over all solute influx from its neighbors. The classical sharp interface model with negligible diffusivity in solid can be described by [40],
(150)$$-V\frac{dc}{dx}=D_{L}\frac{d^{2}c}{dx^{2}},$$
(151)$$V\left(1-k^{e}\right)c^{i}=-D_{L}\frac{dc}{dx},$$
(152)$$T=T_{m}-m^{e}c^{i}-\beta V,$$
(153)$$c\left(\xi^{*}\right)=c_{\infty }.$$

Here, $\xi^{*}\;\mathrm{is}\;\mathrm{denoted}\;\mathrm{as}\;\mathrm{the}\;\mathrm{distance}\;\mathrm{between}\;\mathrm{the}\;\mathrm{solute}\;\sin k\;\mathrm{and}\;\mathrm{the}\;\mathrm{interface}.\;c^{i}$ is the concentration at the interface. The exact solution of Equations (150)–(153) is,
(154)$$c\left(x\right)=c_{\infty }+c_{\infty }\frac{\left(1-k^{e}\right)\left(e^{-\frac{Vx}{D_{L}}}-e^{-\frac{V\xi^{*}}{D_{L}}}\right)}{1-\left(1-k^{e}\right)\left(1-e^{-\frac{V\xi^{*}}{D_{L}}}\right)},$$

Then, the interface velocity is determined by,
(155)$$\beta V=T_{m}-T-\frac{m^{e}c_{\infty }}{1-\left(1-k^{e}\right)\left(1-e^{-\frac{V\xi^{*}}{D_{L}}}\right)},$$

In Equation (155) $\xi^{*}\to \infty \;$ implies that interface velocity is positive if solidus temperature $\;T_{Sol}\;$is greater than temperature *T*, here $T_{Sol}=T_{m}-m^{e}c_{\infty }/k^{e}\;.\;\mathrm{Additionally},\;\mathrm{when}\;\xi^{*}$ has a finite value, the interface velocity is positive if the liquidus temperature is greater than the temperature *T*, here $T_{Liq}=T_{m}-m^{e}c_{\infty }$. Again, an exact solution for partitionless solidification is available. In this case, the interface velocity is given by,
(156)$$V=\frac{T_{0}-T}{\beta },$$

$T_{0}$ is the temperature where the free energies of solid and liquid become equal. For computational work, we considered a diluted solution. Equations (9), (25) and (31) are used for our model with $\;q\left(\Upsilon \right)=4\Upsilon^{3}-3\Upsilon^{4}$. The model system was chosen to be Ni-Cu (0.05 mole fraction alloy). The material parameters used for computation are as follows: $D_{S}=1\times 10^{-14}m^{2}/s$, $D_{L}=1\times 10^{-9}m^{2}/s$, $T_{m}=1728.0\;K,\;\;k^{e}=0.7965$, $m^{e}=310.9$, $T_{Sol}=1708.5\;K$, $T_{Liq}=1712.5\;K,\;\;\sigma =0.37\;J/m^{2}$, β=10 Ks/m, the grid size was 1nm and between the interface the phase field vary from 0.05 to 0.95. From Equations (155), (152), (104) and (156) putting the value of $T_{Sol}$ and Equation (126), the relations of interface velocity is expressed as,
(157)$$\beta V=T_{Sol}-T,$$
(158)$$\beta V=T_{Sol}-T+m^{e}\left(\frac{c_{\infty }}{k^{e}}-c^{i}\right),$$
(159)$$\beta V=T_{Sol}-T+m^{e}c_{\infty }\left(\frac{1}{k^{e}}+\frac{lnk^{e}}{1-k^{e}}\right),$$
(160)$$\beta V=T_{Sol}-T+\frac{m^{e}}{k^{e}}\left(c_{\infty }-c_{S}^{i}\right),$$

Here, Equation (157) is for analytical solution for classical sharp interface mode. Equation (158) is the sharp interface model with diffusion in liquid only. Equation (159) is the analytical solution for partitionless solidification, and Equation (160) is for the thin interface limit at low Péclet number conditions.

## 3. Results and Discussion 

Figure 4 shows the variation of interface velocity as a function of undercooling $T_{Sol}-T$ ignoring the solute sink ($\xi^{*}\to \infty )$. The solid straight line shows the analytical solution of Equation (155). The curved dashed line shows the analytical solution of partitionless solidification$\;V=\left(T_{0}-T\right)/\beta$. The dotted line and the green line are calculated from the thin interface limit and sharp interface limit. For a thin interface, limit the interface velocity marge with an analytical solution as undercooling decreases. At large undercooling, the sharp interface limit’s interface velocity is close to the analytical solution of partitionless solidification. 

Figure 5 shows the variation of interface velocity as a function of the distance between interface and solute sink${\;\xi }^{*}$ in liquid neglecting the kinetic coefficient. The system temperature was 1709 K. The curved line shows the analytical solution of Equation (155). The physical meaning of zero kinetic coefficients in thin interface alloy means a decrease of solid composition by phase-field alloy or increase of the solid composition by interface thickness, bringing the solute trapping effect increases the solid composition. 

Figure 6 shows the variation in solid composition along with the interface for Υ<0.5. The vertical axis represents the relative difference between the measured solid composition at the interface $\;c_{S}^{i}$ and the equilibrium composition $c_{S}^{e}$, scaled by the equilibrium composition $c_{S}^{e}$. Figure 7a show the change in variation of solid composition with a coefficient of phase-field gradient energy, $\beta^{S0}$ along with the interface for $\;D^{*}=1$. Figure 7b shows the change in Υ with a coefficient of phase-field gradient energy along with the interface for $D^{*}=1\;$. We did not detect any difference in Deviation of solid composition and Υ profile for different $D^{*}$ values. From the plot, we can see that as the coefficient of gradient energy decreases, the phase-field model approaches the sharp interface limit. Figure 8a,b show the deviation of solid composition at the interface with diffusivity: $D^{*}=1,D^{*}=2\;\mathrm{and}\;D^{*}=5\;\mathrm{for}\;\beta^{S0}=1\;\mathrm{and}\;\beta^{S0}=2,\;\mathrm{respectively}$. From the figure, we can say that diffusivity has no effect on solid composition in the interface region.

For the validation of our simulation with the analytical result, we use the analytical solution of GL equation Equation (74) and simulation result of Υ with the same material properties. Figure 9 shows that the analytical solution coincides with the simulation result. So, the simulation results are in excellent agreement with the analytical solution, indicating the current implementation of the finite element code. 

## 4. Conclusions

We developed a phase-field model for the solidification of binary alloys, which satisfies the stability conditions at all temperatures. The proposed phase-field potential is composed of local and gradient energy terms, and we derived the GL equations governing the solidification kinetics using the second law of thermodynamics followed by the Onsager assumption. The phase-field model was reformulated for the dilute approximation limit and solved analytically for the 1D problem. We also demonstrate that the proposed phase-field model reduced to the sharp-interface solution at the thin interface limit. 

We developed the relationship between kinetic coefficient and mobility for thin-interface limit, neglecting diffusivity in the solid phase. Our results indicate that the solid composition increases with the effect of finite interface thickness and decreases with finite phase-field mobility. With a zero kinetic coefficient, both these effects are canceled out that results in equilibrium at the interface. 

Using this model for 1D steady-state dilute solution with negligible diffusivity, we observed the concentration profile as a function of Péclet number, which is a function of interface velocity. From the analytical solution, it is concluded that with increasing interface velocity, the concentration profile decreases. The distribution of the partition coefficient was also obtained. From the relation between interface velocity and partition coefficient, we can see that with high interface velocity, the value of partition coefficient goes close to unity for sharp interface model. For 1D dilute solution, we performed numerical simulations. From simulation results, we can conclude that the concentration profile’s height is inversely proportional to the coefficient of gradient energy.

## Figures and Tables

**Figure 1 materials-16-00383-f001:**
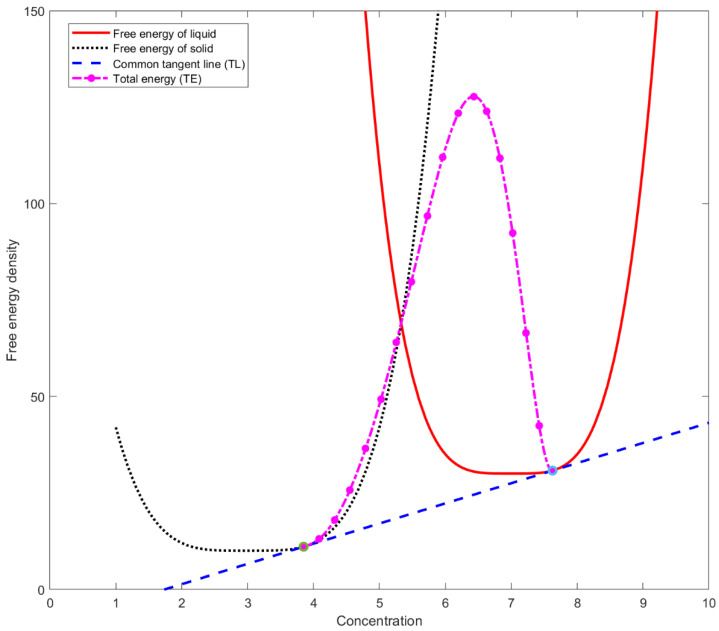
Free energy densities of solid and liquid vs. composition.

**Figure 2 materials-16-00383-f002:**
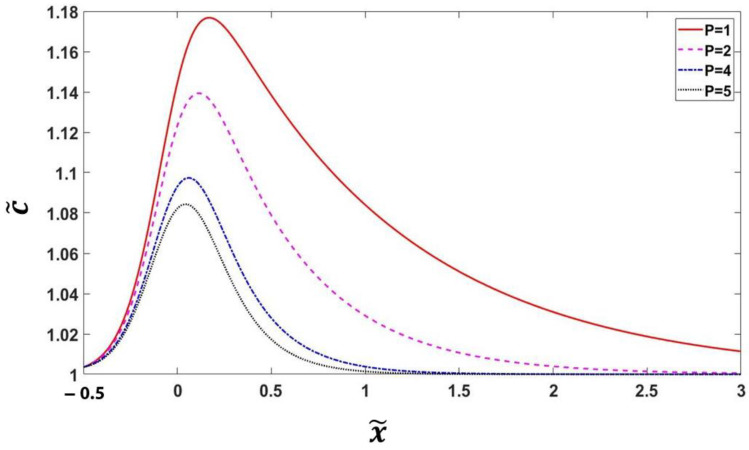
Variation of concentration profile for different Péclet numbers *P*.

**Figure 3 materials-16-00383-f003:**
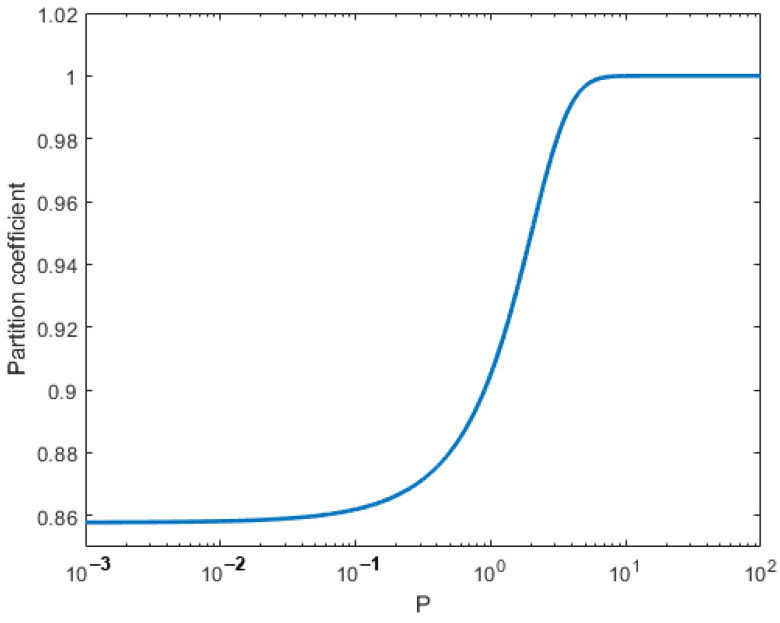
Variation of partition coefficient as a function of *P*.

**Figure 4 materials-16-00383-f004:**
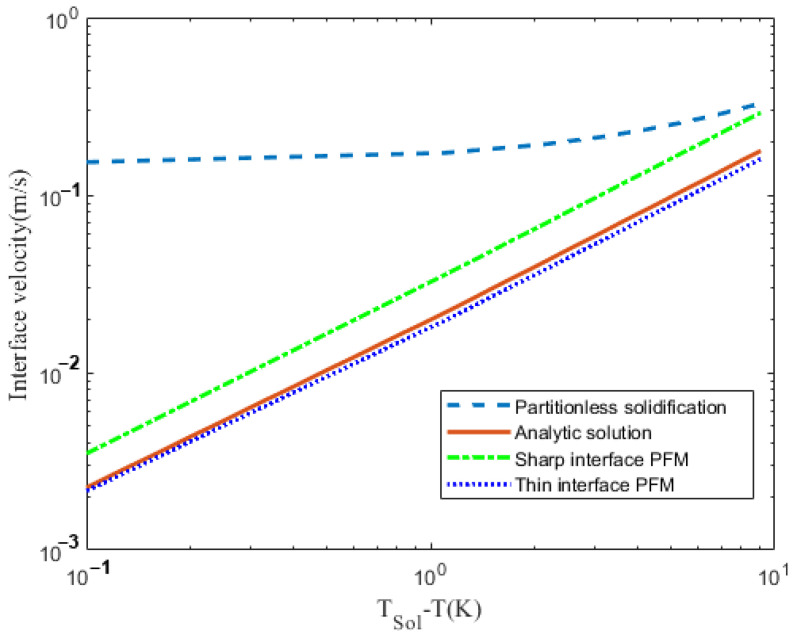
Variation of interface velocity, calculated at $\xi^{*}\to \infty$ as a function of $T_{Sol}-T$.

**Figure 5 materials-16-00383-f005:**
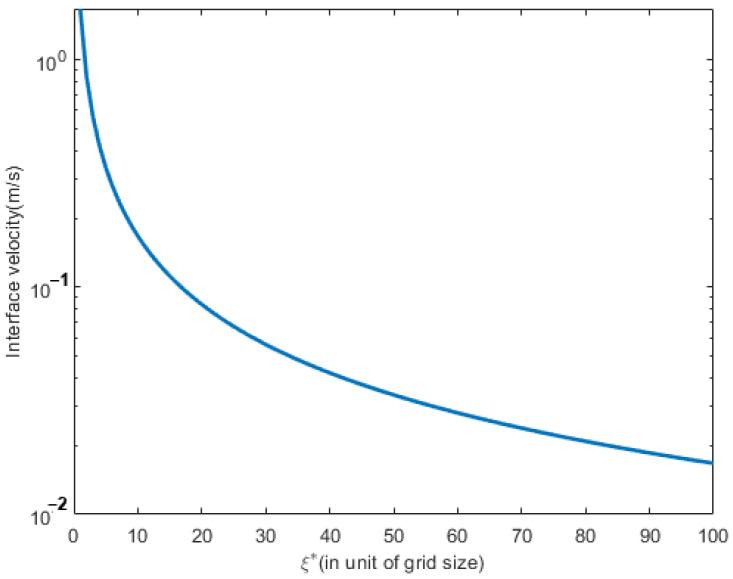
Variation of interface velocity calculated neglecting kinetic effect as a function of $\xi^{*}$.

**Figure 6 materials-16-00383-f006:**
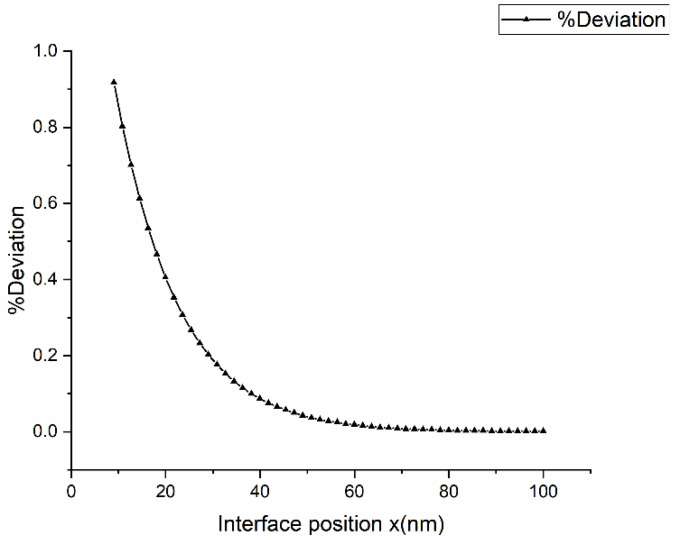
Variation of solid composition along with the interface.

**Figure 7 materials-16-00383-f007:**
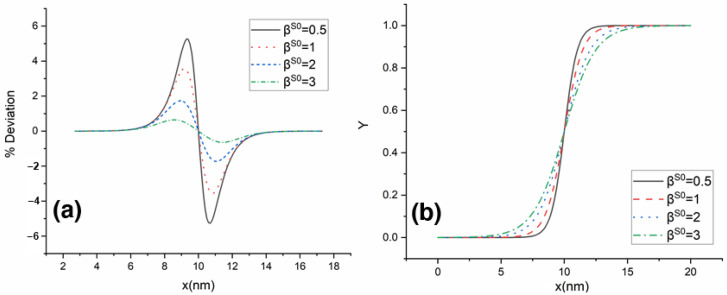
Deviation of solid composition (**a**) and Υ profile (**b**) along the interface for dimensionless diffusivity$\;D^{*}=1$ for various $\beta^{S0}$ values.

**Figure 8 materials-16-00383-f008:**
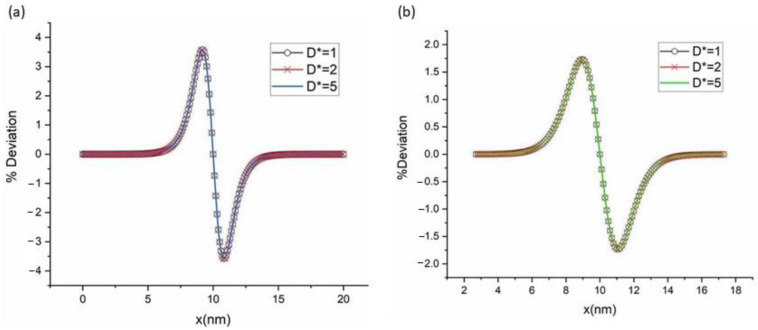
Deviation of solid composition along with the interface with dimensionless diffusivity$\;D^{*}=1,D^{*}=2,\;\mathrm{and}\;D^{*}=5\;$for (**a**) $\beta^{S0}=1;\;$ (**b**) $\beta^{S0}=2$.

**Figure 9 materials-16-00383-f009:**
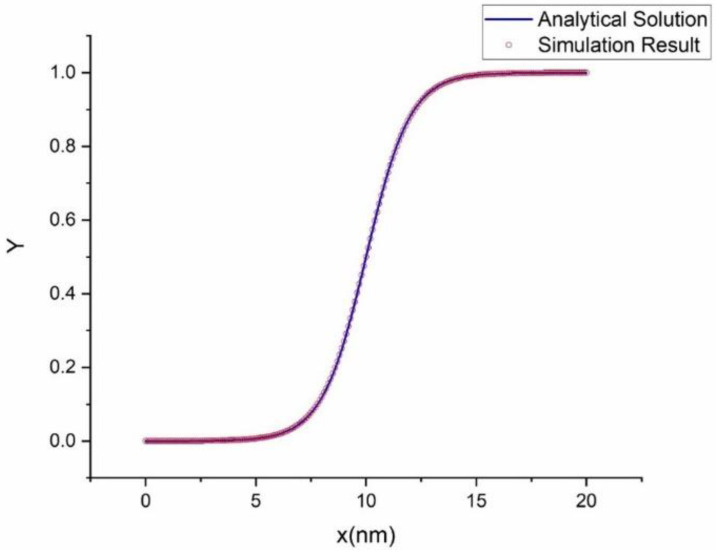
Comparison of numerical result with the analytical result. Our finite element code reproduces matches with the analytical solution.

## Data Availability

Data will be provided upon request.

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
