# Peer review of "A Phase-Field Model for In-Space Manufacturing of Binary Alloys"

_materials, 2022, doi:10.3390/ma16010383_

Round 1

Reviewer 1 Report

Summary:

The manuscript proposes a phase field model to simulate binary alloy solidification with a focus on application to In Space Manufacturing. Using Ni-Cu as an example, the authors demonstrate  the stability of their model under a wide variety of circumstances, including wide temperature ranges.

Notes:

The introduction keys in on in space manufacturing, however the model itself doesn’t have anything specifically related to space that I can see. It is possible I missed that somewhere.

Page 7 line 195:

Where is gamma^l and gamma^s originally defined? This looks very similar to the variable you use for the non-conserved order parameter which makes it difficult to differentiate between them in large math equations.

Page 8 line 216:

Tylor’s should be Taylor’s

Page 25 figure 7:

Images a, b, and c all look the same, images d, e, and f, also all look the same. I don’t see the need for all of them. If you want to show that the diffusivity affects the plots maybe it would be easier to see on the same plot. If you are indicating that D has no effect on the outcome maybe just use figure 8 instead of putting 3 identical images.

Personal preference note:

There are over 11 pages of equations in this paper. Personally I would recommend reducing the number of equations in the paper drastically to improve readability. I would also recommend using variables more commonly used in phase field modeling. In this manuscript, psi is used for free energy which I have never seen in any other paper, typically a combination of F and f is used to represent the different pieces of a free energy function. Also, the non-conserved order parameter used in this paper is gamma, which is usually used to represent interface energy. I would recommend using eta or psi for the phase variable to avoid this confusion. Combining the rarely used variable terms with the over 100 equations which are already complex makes the math difficult to follow and the paper extremely difficult to read. That said, I do believe the math looks correct and well done.

Author Response

Response Letter (materials-2098026)

We appreciate the reviewers’ time and effort for carefully reading our manuscript, and their helpful suggestions and comments. Modifications have been made accordingly, highlighted in yellow in the revised manuscript. The following are our responses (colored) to the reviewer’s comments (in italic) and corresponding modifications to the manuscript (underlined):

Reviewer 1

The manuscript proposes a phase field model to simulate binary alloy solidification with a focus on application to In Space Manufacturing. Using Ni-Cu as an example, the authors demonstrate  the stability of their model under a wide variety of circumstances, including wide temperature ranges.

  • The introduction keys in on in space manufacturing, however the model itself doesn’t have anything specifically related to space that I can see. It is possible I missed that somewhere.

We would like to thank Reviewer 1 for this valuable comment. The model was developed considering the physics governing the FDM process, which is the method of choice for ISM, considering bimetallic feedstock. Although can be used to study other processes that follow the same physics.

  • Page 7 line 195: Where is gamma^l and gamma^s originally defined? This looks very similar to the variable you use for the non-conserved order parameter which makes it difficult to differentiate between them in large math equations.

We would like to thank Reviewer 1 for this valuable comment. They are defined in line 186 on page 7. We added an extra description to clarify, i.e.,

 are the activity coefficients of solid and liquid phases, respectively. They are a measure of how much the thermodynamic characteristics of that mixture deviate from those of the ideal mixture.”

  • Page 8 line 216: Tylor’s should be Taylor’s

We would like to thank Reviewer 1 for this priceless comment and updated the text accordingly.

  • Page 25 figure 7:Images a, b, and c all look the same, images d, e, and f, also all look the same. I don’t see the need for all of them. If you want to show that the diffusivity affects the plots maybe it would be easier to see on the same plot. If you are indicating that D has no effect on the outcome maybe just use figure 8 instead of putting 3 identical images.

We would like to thank Reviewer 1 for this comment. We updated Figure 7 and the text accordingly. We only presented the results for D*=1 and in the main text stated that the results were independent of D*.

  • Personal preference note: There are over 11 pages of equations in this paper. Personally I would recommend reducing the number of equations in the paper drastically to improve readability. I would also recommend using variables more commonly used in phase field modeling. In this manuscript, psi is used for free energy which I have never seen in any other paper, typically a combination of F and f is used to represent the different pieces of a free energy function. Also, the non-conserved order parameter used in this paper is gamma, which is usually used to represent interface energy. I would recommend using eta or psi for the phase variable to avoid this confusion. Combining the rarely used variable terms with the over 100 equations which are already complex makes the math difficult to follow and the paper extremely difficult to read. That said, I do believe the math looks correct and well done.

We would like to thank Reviewer 1 for this valuable and supportive comment. We decided to present details of mathematical derivatives to facilitate reproducibility of the results. The set of variables chosen here was following our previous papers and other phase field papers in the mechanochemistry community. We will adopt the suggested naming convention in our future papers.

Reviewer 2 Report

In this manuscript, the authors proposed a phase-field model for the solidification of binary alloys. However, the novelty of the work is not apparent in terms advantages of the proposed model compared with the existing models and applications for In-Space Manufacturing. In particular, 160 formulations in the manuscript are hard to understand and not necessary at all and therefore require reorganization for ease of following. Moreover, there are no numerical examples regarding In-Space manufacturing.

In light of the concerns above, the current version is not recommended for publication. 

Author Response

We would like to thank Reviewer 2 for the comments.  The key characteristics of the model that differentiates it from previous studies have been elaborated in line 83, i.e., “we develop a phase-field potential for binary alloys that satisfies the stability conditions at all temperatures”.  This feature is specifically crucial for quantitative simulations needed for any particular design.  The proposed model is mainly designed for the additive manufacturing of materials using the FDM process – the most viable process for ISM – for bimetallic feedstocks.  Here, we presented the formulation and their derivatives in extended detail to facilitate the reproducibility of the results by other researchers.  We considered simplified yet scientifically essential case studies that are analytically tractable in sections 2.1 to 2.5 that contain 120 equations, which can be omitted by readers more interested in the numerical implementation of the model.  In the latter case, the total number of equations will reduce to 40.  We updated the text as follows to clarify:

We will consider simplified cases of technological importance in the next four subsections, i.e., secs 2.1 to 2.5, which are included for completeness and can be omitted if numerical simulations of the model are of interest.

Numerical simulations were performed in Sec 2.6 for the Ni-Cu, two critical elements for the printing of aerospace components and functionally graded materials with aerospace applications.

Round 2

Reviewer 1 Report

Looks better.